# Stretchable surface electromyography electrode array patch for tendon location and muscle injury prevention

Shuaijian Yang [1,2], Jinhao Cheng[1], Jin Shang[1], Chen Hang[1], Jie Qi[1], Leni Zhong[1], Qingyan Rao [1], Lei He[1], Chenqi Liu[1], Li Ding[1], Mingming Zhang [1], Samit Chakrabarty [2] ✉ & Xingyu Jiang [1] ✉

Surface electromyography (sEMG) can provide multiplexed information about muscle performance. If current sEMG electrodes are stretchable, arrayed, and able to be used multiple times, they would offer adequate high-quality data for continuous monitoring. The lack of these properties delays the widespread use of sEMG in clinics and in everyday life. Here, we address these constraints by design of an adhesive dry electrode using tannic acid, polyvinyl alcohol, and PEDOT:PSS (TPP). The TPP electrode offers superior stretchability (~200%) and adhesiveness (0.58 N/cm) compared to current electrodes, ensuring stable and long-term contact with the skin for recording (>20 dB; >5 days). In addition, we developed a metal-polymer electrode array patch (MEAP) comprising liquid metal (LM) circuits and TPP electrodes. The MEAP demonstrated better conformability than commercial arrays, resulting in higher signal-to-noise ratio and more stable recordings during muscle movements. Manufactured using scalable screen-printing, these MEAPs feature a completely stretchable material and array architecture, enabling real-time monitoring of muscle stress, fatigue, and tendon displacement. Their potential to reduce muscle and tendon injuries and enhance performance in daily exercise and professional sports holds great promise.

Electrophysiological recordings like electroencephalography (EEG), electrocardiography (ECG) and electromyography (EMG) are routinely used for clinical diagnosis and management of diseases affecting the function of the brain, heart, nerves and muscles[1–5]. With a growth in the field of human-machine interface, sEMG has increasingly become invaluable for non-invasive diagnosis, treatment, daily-life health monitoring and even for control of machines[6–10]. This technology is also useful in sports and trainings. Without appropriate supervision or guide, both amateur and professional athletes can have muscle strain or tendon tear in exercise or trainings[11,12]. Usually, the eccentric contraction can cause muscle and tendon problems,

given that eccentric contractions create more force to muscle and more stretch to tendon than isometric or concentric contractions[13,14]. To prevent muscle injuries, sEMG is a suitable measure because it can show muscle activities with lots of quantified information such as root-mean-square (RMS) and median frequency which are related to muscle condition[2,15,16]. There are also many works studied on neuromuscular junctions by high density sEMG, to demonstrate the muscle fatigue and pain[17–21]. Furthermore, it would be much helpful for injury prevention if the muscle-tendon junction location can be monitored during the exercise. However, there is very little research using sEMG techniques to make such tendon identifications.

[1]Guangdong Provincial Key Laboratory of Advanced Biomaterials, Department of Biomedical Engineering, Southern University of Science and Technology, Shenzhen, Guangdong 518055, P. R. China. [2]School of Biomedical Sciences, Faculty of Biological Sciences, University of Leeds, Leeds LS2 9JT, UK. ✉e-mail: S.Chakrabarty@leeds.ac.uk; jiang@sustech.edu.cn

Electrophysiology involves either recording of electrical signals or means of affecting the signals by delivery of a charge as a stimulus using the electrode. The electrode as interface between human and device is crucial for both techniques above. To achieve accurate data from muscles, the electrode needs to satisfy plenty of requirements, e.g., low impedance for signal quality[22], good stretchability and adhesiveness for signal stability[23], small size (diameter ≤ 5 mm)[16] for high-density recording, good biocompatibility and durability for wearable device. Among these, the greatest one is that most electrodes are rigid which cannot well conform to human skin. This usually causes position shift affecting signal acceptance or electrode failure due to long-term strain of the electrode itself[24].

Currently, existing electrodes have a few limitations. The non-polarizable property helps Ag/AgCl electrode replace traditional electrodes like platinum or gold to be the most used type of electrode in clinic and research. With the usage of conductive gel between electrode and skin, high signal-to-noise ratio (SNR) can be obtained. But the signal quality is usually reduced as the gels desiccate, and in this case, the signal quality can be even worse because of electrode's lack of conformability on skin[25]. Besides, Ag/AgCl electrodes barely have a robust structure, where AgCl film can chip away after repeated use. It also releases highly toxic metal ions into the tissue and has a risk of infection[26]. While devices based on gold or platinum often have flexibility or fatigue issues which restrict their working performances on soft skins[27–31]; besides, lack of adhesiveness at electrode site would also cause motion artifacts in sEMG signals[32]. Thus, developing an adhesive, conformal and non-toxic dry electrode has been the goal of many researchers.

Considering so many demands on electrophysiological electrodes, materials which have excellent flexibility and stretchability, for example, liquid metals hold huge advantage in this field[33–42]. In the previous work in our group, the printable metal-polymer conductor (MPC) made by eutectic gallium indium alloy (EGaIn) is highly stretchable and flexible (strain of 500%), conductive ($8 \times 10^3$ S/cm) and biocompatible, which has been used as flexible circuits, strain sensors and electroporation electrodes[43–48]. Thanks to this flexible circuit, fully conformal multi-channel sEMG electrodes for high-density recording become possible on elastic substrates. But using this liquid metal as the contact electrodes on skin can have leakage and abrasion problems which severely reduce device working time. As a substitute material, poly(3,4-ethylenedioxythiophene) polystyrene sulfonate (PEDOT:PSS) is a more suitable conductive polymer due to its good biocompatibility, high conductivity, and high electrochemical stability[49–51]. However, pure PEDOT:PSS film only has a strain of ~5%, which is not enough for most skin deformation[52]. Its water-soluble property constrains application with sweat and its poor ability against rubbing affects durability when touching the skin[53]. Non-adhesiveness of pure PEDOT:PSS film also has contact issue, thus reducing the electrophysiological signal quality. As a result, how to fabricate a conformal, adhesive and robust dry electrode becomes an issue to address. Dry electrodes force the material to be classified into three types: metal, carbon materials, and conductive polymers. Conventional metal and carbon materials have exceedingly high Young's moduli, which must be fabricated into micro-/nano-structures using complicated procedures for flexible bioelectronics. In comparison, employing conductive polymers that can be variably tuned becomes a more advantageous method since researchers may provide them specific functionalities depending on the application scenarios.

In this work, we reported an adhesive, stretchable, biocompatible and gel-free TPP electrode for sEMG recording based on tannic acid (TA), PEDOT:PSS and polyvinyl alcohol (PVA). Due to its high stretchability and adhesiveness, the electrode can maintain stable contact even in skin folds to give low interface impedance. As a dry electrode, it can have longer working time which is 16 times of Ag/AgCl electrode. Based on MPC circuit, a multi-channel sEMG metal-polymer electrode array patch is fabricated to achieve high-quality and high-density sEMG signals for monitoring of muscle loading and muscle fatigue, whose performance is more stable than commercial sEMG array. More importantly, tendon displacement is continuously monitored by MEAP to control the tendon stretch in a safe range, thus reducing risks of muscle or tendon injuries.

## Results
### Composition and fabrication
The MEAP is formed by two parts: circuit and electrode (Fig. 1a). The material for the circuit part is MPC which has excellent stretchability and conductivity to secure durability of device and quality of signal. Yet MPC is not suitable for contact electrodes because it is not rub-resistant and easy to leak when applied to skin. So, the material for the electrode part is based on PEDOT:PSS. Unfortunately, the strain >5% on a pure PEDOT:PSS film can cause cracks on epidermal electrodes during movements of muscles. In this case, the addition of PVA can raise the flexibility of PEDOT:PSS film visibly, giving the electrode sufficient conformability on the skin to work for longer time. In addition, doping with TA gives adhesiveness to the film (Fig. 1b) and further improves stretchability and conformability of the electrode (Fig. 1c). MEAP can be fabricated within 30 min with simple instruments like screen printers and ovens (Fig. 1d). With the encapsulation of polyethylene glycol blended polydimethylsiloxane based adhesive (PPA) by the previous work in our group[54], the liquid metal circuit is protected and only TPP electrodes attach to the skin, working satisfactorily with high SNR (>20 dB) for 5 days and even longer. With scalable screen-printing fabrication, MEAP is easy to be redesigned for different muscles as required. We can store MEAP on a release film to protect the device; the MEAP also shows excellent flexibility and stretchability whose strain can be over 200% (Supplementary Fig. 1, Supplementary Movie 1); the easy-to-use property and stretchability enable MEAP to be directly attached on the skin to record sEMG signals (Fig. 1e).

### Topography and characterization
To determine the optimal concentration of each component in TPP solution, we first explored the optimal PVA loading in PEDOT:PSS solution, followed by measuring the different properties of PEDOT:PSS film versus PVA doping concentration. Weight fractions of PVA of 0, 25, 33.3, 50, 66.7 and 75 wt% were studied. The stretchability of PEDOT-PVA film improved with increasing concentration of PVA, while the conductivity was the highest when PVA content was 25 wt% (Fig. 2a). Abrasion-resistance tests to study the durability of PEDOT-PVA electrodes were performed as well (Fig. 2b, Supplementary Fig. 2). In addition, electrode-skin impedance, water-resistance and solution viscosities were also tested for selection of the optimal concentration of the added PVA (Fig. 2c, Supplementary Fig. 3). All PEDOT-PVA electrodes show similar impedances to the Ag/AgCl electrode with the same order of magnitude. Accordingly, based on the above results, we concluded that the weight fraction of 66.7 wt%, in other words, weight ratio of 2:1, was the optimal concentration of PVA and PEDOT:PSS, providing higher flexibility and durability suited for application onto the skin (Fig. 2d). Based on this, two types of electrode films: (1) pure PEDOT:PSS, (2) PEDOT:PSS with 66.7 wt% of PVA were chosen for further topographic characterizations using scanning electron microscopy (SEM) and atomic force microscopy (AFM) (Supplementary Figs. 4 and 5). After the ratio between PEDOT:PSS and PVA was determined, different TPP films were made to identify the optimal TA concentration. A few articles have reported that by addition of a certain amount of TA into hydrophilic polymer solution, tacky and elastic supramolecular complexes can be formed, in which TA is the binder between the polymer chains (Fig. 2e)[55–57]. TA contains gallol groups that can form hydrogen bonds, electrostatic interactions, hydrophobic interactions and cation-π interactions with different adherends. The tacky complexes can also improve the wetting behaviour on different

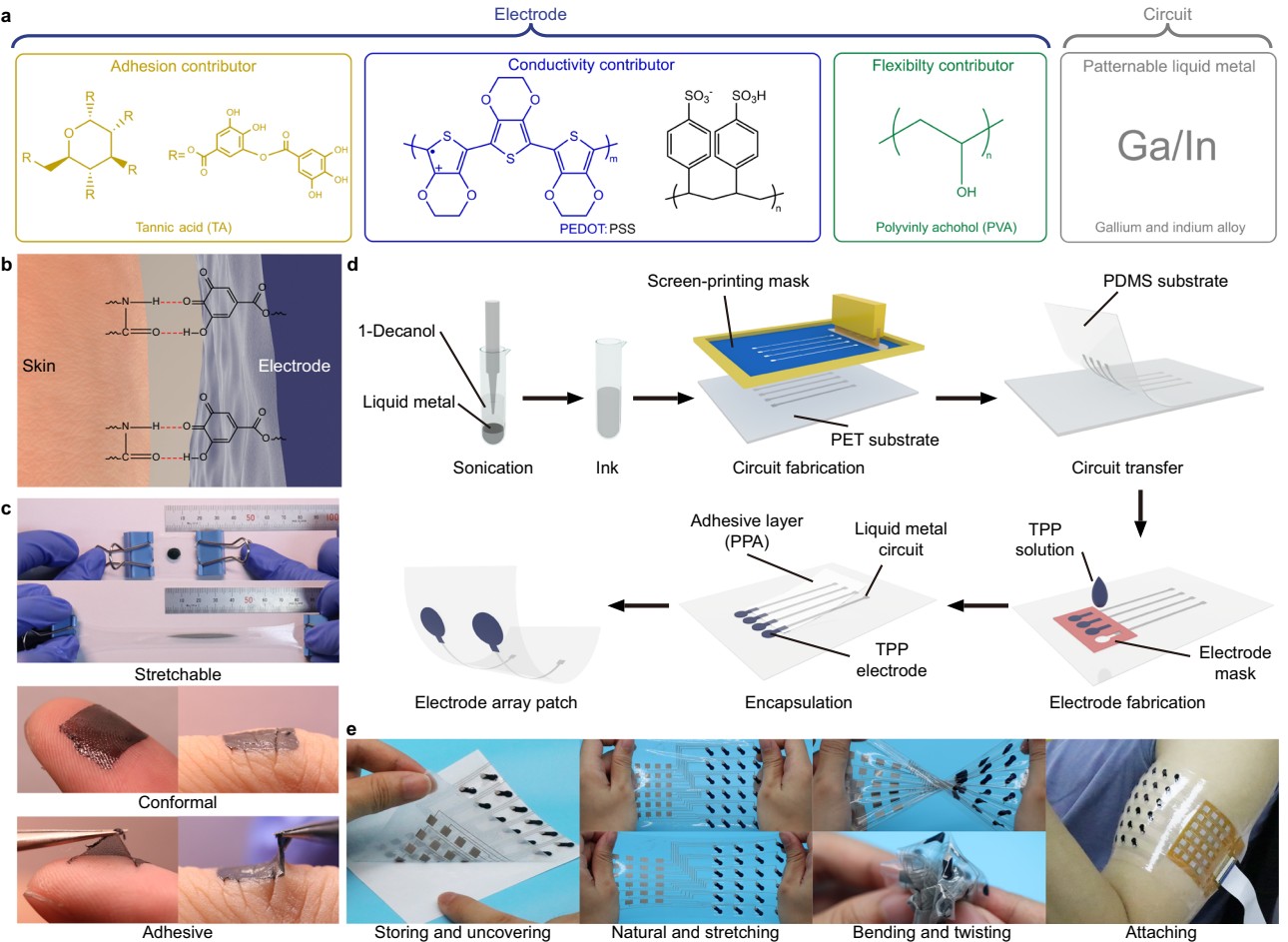

**Fig. 1 | Schematic diagram of the composition and fabrication of MEAP.**
**a** Different constituents and their function in MEAP. **b** Schematic diagram of adhesion between TPP electrode and skin. **c** Photographs of the performance of TPP electrodes during stretching and on skin. **d** The fabrication process of MEAP. The liquid metal ink is prepared to make MPC circuits. TPP contact sites are cast on the substrate using moulds. The adhesive encapsulation layer, PPA, is fabricated to cover circuits and expose contact sites only. **e** Photographs of MEAP during stretching, bending and twisting. MEAP can be stored on a release film and uncovered simply to attach on skin for sEMG recording.

surfaces and form strong adhesion by Van der Waals forces and mechanical interlocks. After the addition of TA, porous structures appear in the film (Fig. 2f and Supplementary Fig. 6). Such porous structures lead to lower cohesive energy and Young's modulus. The decreased cohesive energy of the adhesive electrode may cause cohesive rupture (damage of electrode) rather than adhesive rupture (on the surface) and leave undesirable residue when peeling from the skin (Fig. 2g). Lower Young's modulus gives better compliance and stretchability to the film, which are vital to the conformal adhesion between electrodes and skin[58]. This was also proved by the results of tensile and peeling tests of TPP films with increasing concentration of TA (Fig. 2h, i). All these observations helped us use the final weight concentration of TA at 8%, which makes the film soft and adhesive but not easy to tear. This TPP film shows elongation at break of 188%, Young's modulus of 644 kPa and adhesive forces of 0.58 N/cm on the skin. Once the concentration of the constituents of the TPP solution was determined, each constituent's indispensability was verified by the changes in conductivity, stretchability, and adhesiveness of the electrode (Supplementary Fig. 7). Meanwhile, the TPP film showed good repeatability after being stretched to a strain of 20% for 1000 cycles (Fig. 2j). Furthermore, RMS changes between sEMG signals filtered using two different low cutoffs indicated that, the adhesive TPP electrodes were more stable and able to record the lower frequency effectively even during dynamic tasks (Fig. 2k and Supplementary Fig. 8). This is beneficial as it allows one to reduce the amount of post-

hoc processing prior to analysis, making it adequate to analyse while keeping valuable information as much as possible. The conductivity and electrode-skin impedance of TPP film were also examined (Supplementary Figs. 9, 10). The impedance showed similar performance as the PEDOT-PVA film and was better than the Ag/AgCl within the sEMG frequency range. Further, we found TPP electrodes showed excellent stability in adhesion, skin-electrode impedance and SNR after 200 times of compress or stretch on the skin (Supplementary Figs. 11, 12). TPP electrodes can also be used repetitively without changing the baseline noise (Supplementary Fig. 13). In comparison with dry electrodes in reported literatures[59–67], TPP electrode performs better when the conformability and signal quality are evaluated (Fig. 2l, Supplementary Tables 1, 2). In the comparison, we also found only 6 out of 13 studies discussed adhesive electrodes, and our TPP electrodes perform the best in terms of adhesiveness, which is an important contribution to its highest SNR among all dry electrodes.

## Biocompatibility and durability
The biocompatibility of TPP electrodes was tested on the skin making comparisons with Ag/AgCl electrode (Fig. 3a, b). Both electrodes were biocompatible on the skin and can be worn for 24 h without any itchiness or inflammation. Based on the outcome of biocompatibility test, the long-term test was carried out to identify potential of these electrodes for use in wearables (Fig. 3c). TPP electrodes were placed on the muscle belly of flexor carpi ulnaris (FCU) of a subject. The

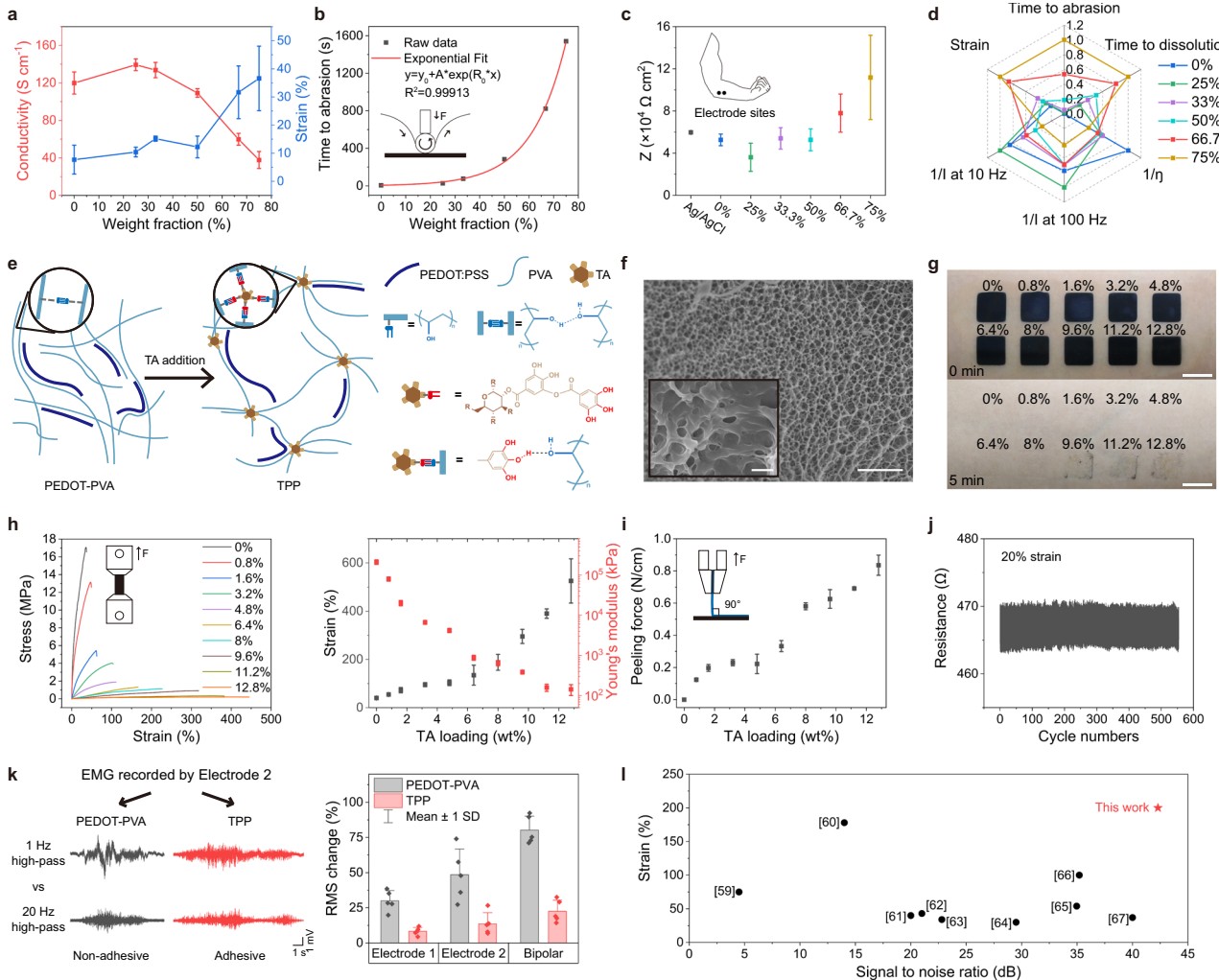

**Fig. 2 | Characterization of PEDOT-PVA and TPP electrodes. a** Strain and conductivity of PEDOT-PVA electrodes. **b** Abrasion-resistance of PEDOT-PVA electrodes. **c** The impedances of the Ag/AgCl electrode and PEDOT-PVA electrodes measured at 100 Hz, as recordings up to 100 Hz is commonly used for comparison of muscle activity. **d** The radar plot of characterization results with different PVA addition. **e** Schematic illustration of the working mechanism in PEDOT-PVA and TPP films and related chemical structures. After TA addition, the PVA chains are more cross-linked at the position of TA. This process expands the space between polymer chains and generates a porous structure. **f** The cross-sectional SEM image of TPP films with 8% TA loading, with a magnified region of interest as inset. Scale bar: 4 μm; inset: 1 μm. Micrographs (×3) were collected independently with similar results.

**g** A photograph of residue after TPP films with different TA loading peeled off from the skin. Scale Bar: 1 cm. **h** Tensile stress–strain curves, strain and Young's modulus of TPP films. **i** Peeling force of TPP films on the skin. **j** Real-time monitoring of the TPP film by stretching the film from a strain of 0 to 20% for about 500 cycles. **k** EMG signals recorded by PEDOT-PVA and TPP electrodes. Electrodes 1 and 2 were attached to biceps brachii for recording during the biceps curl. The raw data was filtered by 1 Hz and 20 Hz high-pass filter; RMS change between two filtered signals showed the extent of discrepancy. RMS change = $\frac{RMS_{1Hz} - RMS_{20Hz}}{RMS_{1Hz}}$. **l** Strain and SNR comparisons between this work and electrodes in other literatures. Data in Fig. 2 are presented as mean values ± SD. For all measurements, number of samples was 3, except for Fig. 2k it was 5.

results showed TPP electrodes can work superbly holding SNR level above 30 dB for 3 days and above 20 dB for almost 5 days, yet Ag/AgCl electrode decreased to 20 dB within 6 h (Fig. 3d). The total wear-time for long-term test was over 10 days and TPP electrodes caused no itchiness or inflammation. As for the increase of baseline noise, we hypothesize that perspiration in normal life might gradually affect contact area and conductivity of the TPP film because sweat fat and salts can only slowly accumulate on the TPP film, increasing the noise intensity[68]. Furthermore, the natural metabolic processes that lead to an increase in stratum corneum thickness can elevate the impedance between the electrodes and the skin during prolonged measurements. As for the MEAP, we checked the reattachment performance of the patch (Supplementary Fig. 14). The results showed all channels of MEAP have stable performances. This indicates that MEAP can be used repetitively. We also examined the permeability performance of MEAP for daily long-term use

(Supplementary Fig. 15). The results demonstrate that the permeability of the MEAP is well-suited for extended periods of usage, as it does not hinder the normal evaporation of sweat from the skin. Furthermore, we discovered that the permeability of the MEAP can be adjusted by modifying the physical structure of the substrate. This ability to tune the permeability enables us to create a comfortable wearing experience for daily use, as the permeability can be increased to a level that promotes adequate airflow. After that, we cultured Human Umbilical Vein Endothelial Cells (HUVEC) to further demonstrate the biocompatibility of the device. Fluorescent staining images showed a regular cell morphology and almost no dead cells (red) in both groups, indicating the encapsulation layer (PPA) and electrode contact sites (TPP) are not toxic to the live cells (green) (Fig. 3e). The cell viability tests were also performed on PEDOT-PVA, LM and PDMS (Supplementary Fig. 16), showing that the cell compatibility for all materials is good.

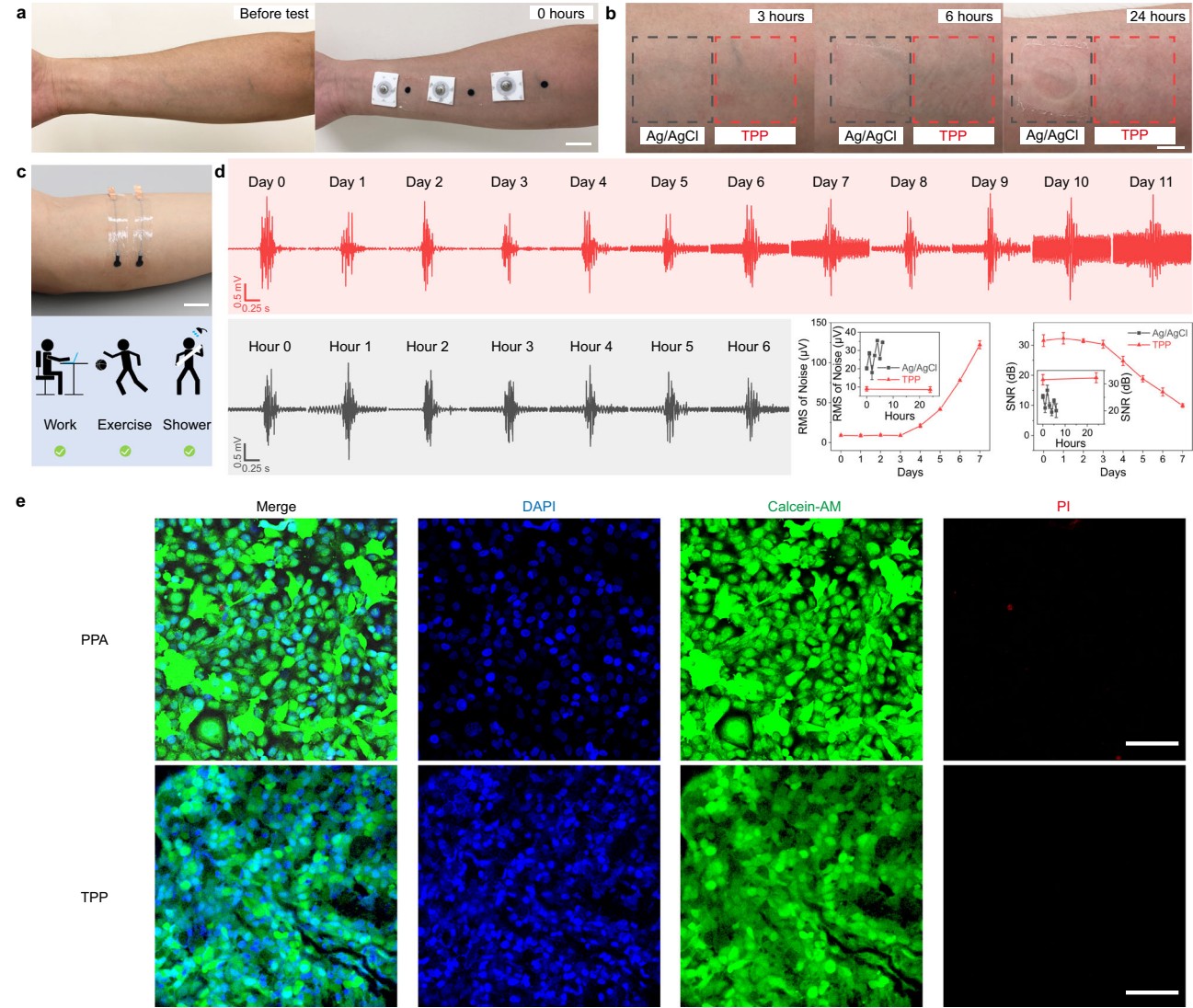

**Fig. 3 | Biocompatibility and long-term test of TPP electrodes. a** Photographs of arm skin of the subject for the biocompatibility test. Scale bar: 2 cm. **b** Skin condition after wearing of 3, 6, and 24 h. Scale bar: 1 cm. After 6 hours' attachment, glue came off from Ag/AgCl electrodes and sticked tightly on the skin, which was very hard to wash off. However, with TPP electrode, the PPA shows very little glue left on the skin after 24 hours' adhesion. **c** A photograph of the electrodes for long-term test, and a schematic diagram to illustrate activities of the subject during the test. Scale bar: 2 cm. **d** sEMG signals recorded by TPP and Ag/AgCl electrodes during the long-term test ($n = 3$ samples for each measurement). Data are presented as mean values ± SD. **e** Fluorescent images of endothelial cells grown on PPA and TPP films to show their biocompatibility. Scale bar: 100 μm. Micrographs (×3) were collected independently with similar results.

## Flexibility and conformability of TPP electrodes

The good contact between electrode and skin can guarantee the stability and quality of signals, especially when recording on positions with skin folds. To verify this point, we recorded sEMG signals on frontalis muscle by Ag/AgCl and TPP electrodes. The sEMG of frontalis muscle is useful in research and treatment of sleep disorders, anxiety, headaches, facial recognition and so on[69]. But skin folds are common on the frontalis muscle which cause difficulty for non-conformal electrodes to record sEMG signals. Due to the excellent flexibility and adhesiveness of electrode and substrate, TPP electrodes can always make perfect attachment to the skin no matter if the skin is compressed or stretched (Fig. 4a). During recording, the TPP electrode fit in the skin folds, providing stable signals when frontalis muscle contracted while the Ag/AgCl electrode failed after only 4 contractions (Fig. 4b). In the case of non-conformal electrodes, these skin folds would appear when frontalis muscle contracted, creating air gaps between electrodes and skin (Fig. 4c). These air gaps decreased adhesion force between electrode and skin, making electrodes easy to

fall off. Meanwhile, the air gaps introduced more noise which is demonstrated by a circuit model (Supplementary Fig. 17). By contrast, the TPP electrode kept the same shape with skin folds to provide stable contact during muscle contraction. After quantification, we found that the TPP electrode was much more stable than the Ag/AgCl electrode for sEMG recording over skin folds, based on RMS of noise level and SNR (Fig. 4d). Another important application of sEMG in clinical diagnosis is to evaluate muscle function according to the median frequency of signal[70,71]. Usually, the decrease of median frequency indicates the fatigue of muscle; but if the decrease shows up at the beginning of test, it means the muscle has pathological change which can only recruit few of fibres for the contraction[71,72]. To assess the ability of TPP electrodes to obtain information of frequency in the signal, Ag/AgCl and TPP electrodes were set on the same position on FCU (Fig. 4e). The TPP electrodes showed a little better SNR than the Ag/AgCl electrodes that they are 39.2, 37.5 and 40.5 dB for three contractions recorded by TPP electrodes and 38.9, 37.5 and 38.6 dB by Ag/AgCl electrodes. The spectrograms showed TPP electrodes can

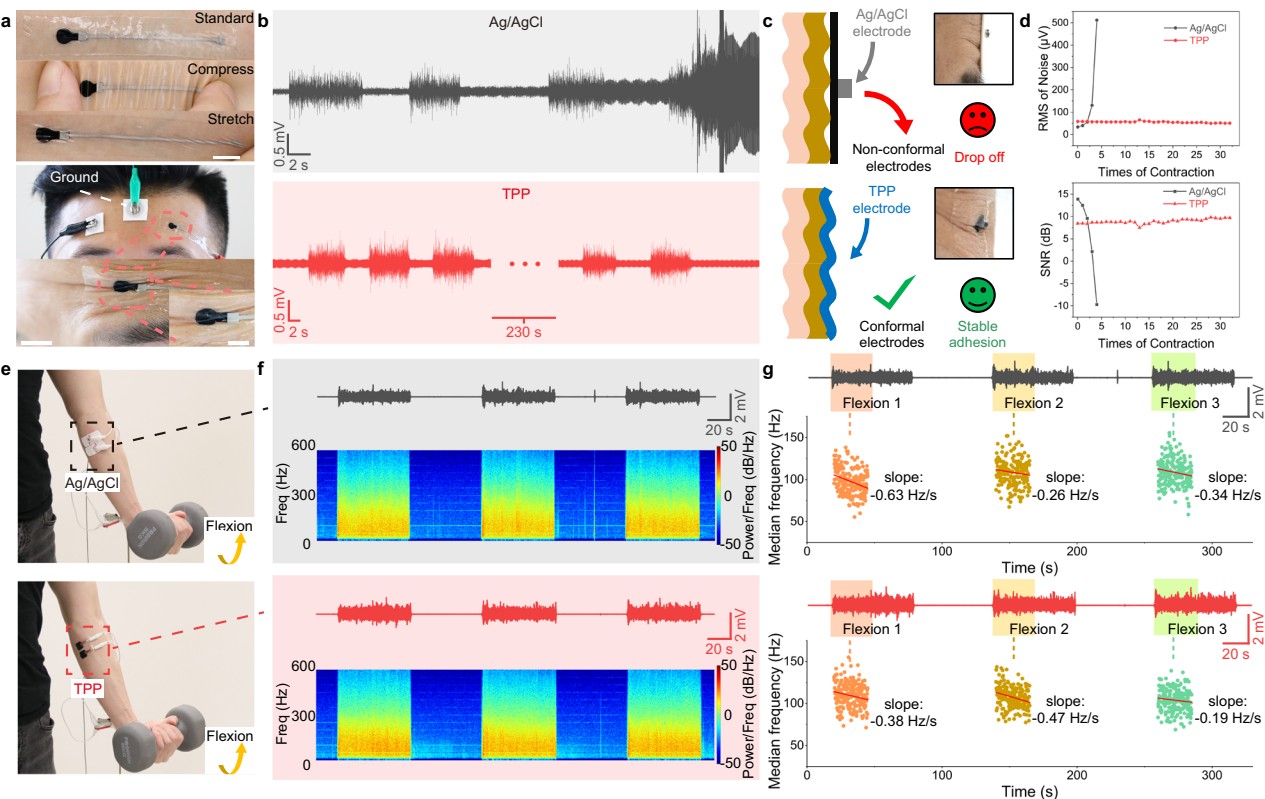

**Fig. 4 | Comparison of recording performances on skin between Ag/AgCl and TPP electrodes. a** Top, standard, compressing and stretching TPP electrodes on the skin. Scale bar: 1 cm; bottom, photographs of Ag/AgCl and TPP electrodes when recording sEMG of frontalis and the TPP electrode in the skin folds. Scale bar of photographs at the bottom: 1 cm; bottom inset: 0.5 cm. **b** sEMG signals recorded by Ag/AgCl and TPP electrodes, respectively. The subject was asked to make each contraction for 5 seconds. In the case of recording by Ag/AgCl electrodes, after four times of contraction, noises were even higher than signals. **c** Schematic illustrations and lateral photographs of Ag/AgCl electrode and TPP electrode on skin folds.

**d** Noise level and SNR recorded by two electrodes during contractions. **e** Photographs of electrode configuration on FCU and contraction task. Two pairs of electrodes were attached to the same position on the forearm. **f** sEMG signals and spectrograms recorded by Ag/AgCl and TPP electrodes, respectively. The subject was asked to curl the wrist with a 5 kg dumbbell for three sessions to activate FCU. **g** sEMG signals and linear fit results of median frequency during flexion 1, 2 and 3 recorded by Ag/AgCl and TPP electrodes. Decreasing median frequencies indicated fatigue of the muscle.

give clear frequency information just like Ag/AgCl electrodes (Fig. 4f). To compare the performances of Ag/AgCl and TPP electrodes on fatigue measurement, the subject was asked to curl the wrist 60 s for three times for each type of electrodes. Three tasks were named as flexion 1, 2 and 3 to calculate median frequency during each task (Fig. 4g). To reduce errors caused by fatigue, we linear fitted the first 25 s of each contraction to quantify the change[73]. The slopes obtained by two types of electrodes both showed negative values which indicated the muscle was in fatigue. This test proved the TPP electrodes can measure the muscle fatigue the same as Ag/AgCl electrodes. Apart from similar abilities to record different arm positions (Supplementary Fig. 18), these results show TPP electrode can record accurate frequency information of sEMG signals, which is even more reliable than Ag/AgCl electrodes.

**Comparison between MEAP and commercial sEMG array**

Electrode arrays which adhere to the skin have been developed and used in laboratory settings. The material for the most popular sEMG commercial array (CA) currently available is polyimide (PI), which has the Young's modulus of 3 GPa. Due to its characteristics, it was found that unless specific features, such as serpentine design, were introduced, the array with PI substrate cannot make a fully conformal contact with human skin (Young's modulus of 10 kPa). To fairly compare the contact performance on the skin, we fabricated a 64-channel MEAP with the same configuration as the CA (Supplementary Fig. 19). We recorded a movie to contrast the contact effectiveness of the PI

sEMG commercial array and the MEAP on the biceps brachii (Fig. 5a, Supplementary Movie 2). CA formed gaps between itself and the skin during the muscle action, yet the MEAP always maintained great contact, even though their thicknesses are both 100 μm. The MEAP was connected to the sEMG recording system via flexible printed circuit (Supplementary Fig. 20). The differences in attachment performance were reflected directly in the sEMG signals. When CA was applied to the muscle, the poor attachment caused gaps between electrodes and the skin after the muscle contractions, resulting in the increase in baseline noise during the rest stage, which would lower the SNR level (Fig. 5b). To demonstrate the effect of skin deformation on the recording performance, we calculated the SNR of the first and the last contractions. We recorded and displayed the SNR of each channel, and none of the CA electrodes provided SNR >20 dB with the last contraction, whereas all MEAP channels provided SNR >20 dB for both the first and last contractions (Fig. 5c). We believe that the conformal attachment is the determining factor in this because the SNR of the first contraction recorded by CA was adequate but significantly worsened for the last contraction. We also employed statistical analysis to quantify these outcomes (Fig. 5d). Because of the mismatch between CA and the skin, the baseline noise level of all CA electrodes rose distinctly after only one contraction. After reattachment, the baseline noise was lowered, indicating again that the mismatch between CA and skin is the cause for change. Most CA channels showed more than two-fold change in baseline noise, resulting in a significantly lower SNR. While the MEAP exhibited a much more stable noise level even after ten contractions,

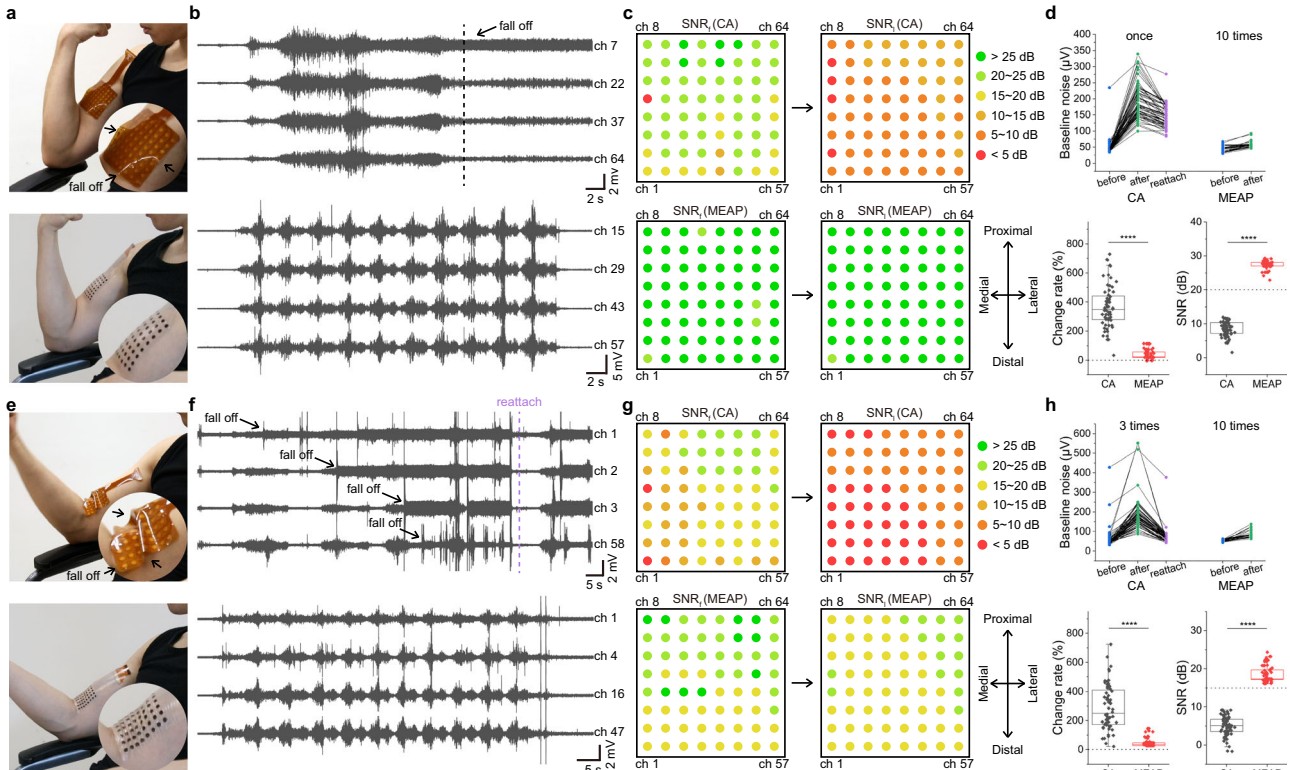

**Fig. 5 | The comparison of attachment performances between commercial array and MEAP. a, e** Photographs of CA and MEAP attachment, illustrating the difference when recording from muscle and muscle-tendon junction of biceps brachii. **b, f** sEMG signals recorded using CA and MEAP on muscle and muscle-tendon junction of biceps brachii. Four typical channels were picked for each recording. **c, g** Spatial SNR performance map for each channel of CA and MEAP for the first and last muscle contraction. $SNR_f$: SNR of the first contraction; $SNR_l$: SNR of the last contraction. **d, h** Statistical analysis of performances of CA and MEAP, including baseline noise level of CA before and after one or three muscle contractions, as well as after reattachment; baseline noise level of MEAP before and after ten muscle contractions; baseline noise change rates before and after muscle contractions; SNR performance of the last muscle contraction recorded by each of the CA and MEAP channels ($n = 64$ for each measurement). The box plots show the mean (centre square), median (centre line), the 25th to 75th percentiles (box) and the smallest and largest value that is ≤1.5 times the interquartile range (the limits of the lower and upper whiskers, respectively) Significance was determined by one-sided $t$ test (*$P < 0.05$; **$P < 0.01$; ***$P < 0.001$; ****$P < 0.0001$).

maintaining a high SNR. We also recorded sEMG signals from muscle-tendon junctions (closer to the distal end of the biceps) with both arrays, where skin deformation was greater (Fig. 5e, Supplementary Movie 3). The mismatch between CA and skin was significant, while the MEAP still kept perfectly conformal contact. The sEMG signals also revealed a drop-off of signal from some electrodes of CA (Fig. 5f). Recordings from the CA channels on muscle-tendon junction were significantly damaged by muscle contractions, because SNR of most channels decreased from the first contraction to the last one (Fig. 5g). MEAP on the other hand, produced stable recordings with all channels having SNR >15 dB. Statistical analysis of sEMG data from CA on the distal end produced similarly unsatisfactory results, but it is worth noting that MEAPs always kept stable and high-quality recording even after ten muscle contractions (Fig. 5h). These results suggest that the MEAP can record better sEMG signals than CA because of the better contact even with large skin deformation.

## MEAP for sEMG activation mapping
Surface areas of Ag/AgCl electrodes range from 1 - 2 cm² to ensure quality of the signals. The spatial resolution of these electrodes is limited, owing to their surface areas when recording activity of both small and large muscles. An optimal electrode should be lower than 5 mm in size and the inter-electrode distance (IED) should be lower than 10 mm[16]. Using TPP electrodes, we can fit the optimal design. In addition, different dimensions and specific shapes of sEMG electrodes can be fabricated based on the recording needs (Fig. 6a). Here, we developed MEAPs for recording sEMG as the biceps brachii (BB) and

abductor pollicis brevis (APB) contracting (Fig. 6b). MEAPs can be conformally attached on the skin over BB and APB either during rest or contraction and 24 contact sites were labelled as A1 to D6 (Fig. 6c, d). The recorded signals showed good SNR and high stability (Fig. 6e, f) owing to the device's stretchability. Muscle activity maps based on RMS values were generated to visualize the advantage of high-density systems. For BB recording, there was no activity prior to the task (Fig. 6g). At the start of curling, the activity was recorded at all the electrode sites, with the long head more active than the short head of BB. When BB was fully contracted, the fibres were shortened and the muscle belly was farther from the distal end, which caused the active zone to move to the right in the muscle activity maps with an increased activity. When the subject performed eccentric contraction, lowering the dumbbell back to rest position, there was lower muscle activity than in the concentric contraction, and the active zone moved back to the middle (Supplementary Movie 4). Owing to the array design, the entire process of muscle movement underneath could be effectively monitored. Similarly, when recording on APB, the recruitment was along the diagonal of the MEAP (Fig. 6h). Unlike in BB, there was no shift of activation zone in APB. Compared to BB, recordings of APB by the MEAP with smaller electrodes and shorter IED showed lower activity but finer recruitment. These results indicated that recruitment and length change in muscle during contractions can be recorded accurately by MEAPs with high spatial and time resolution. Such tools will create remarkable benefits and provide a potential means for clinical diagnosis, medical treatment and sports sciences.

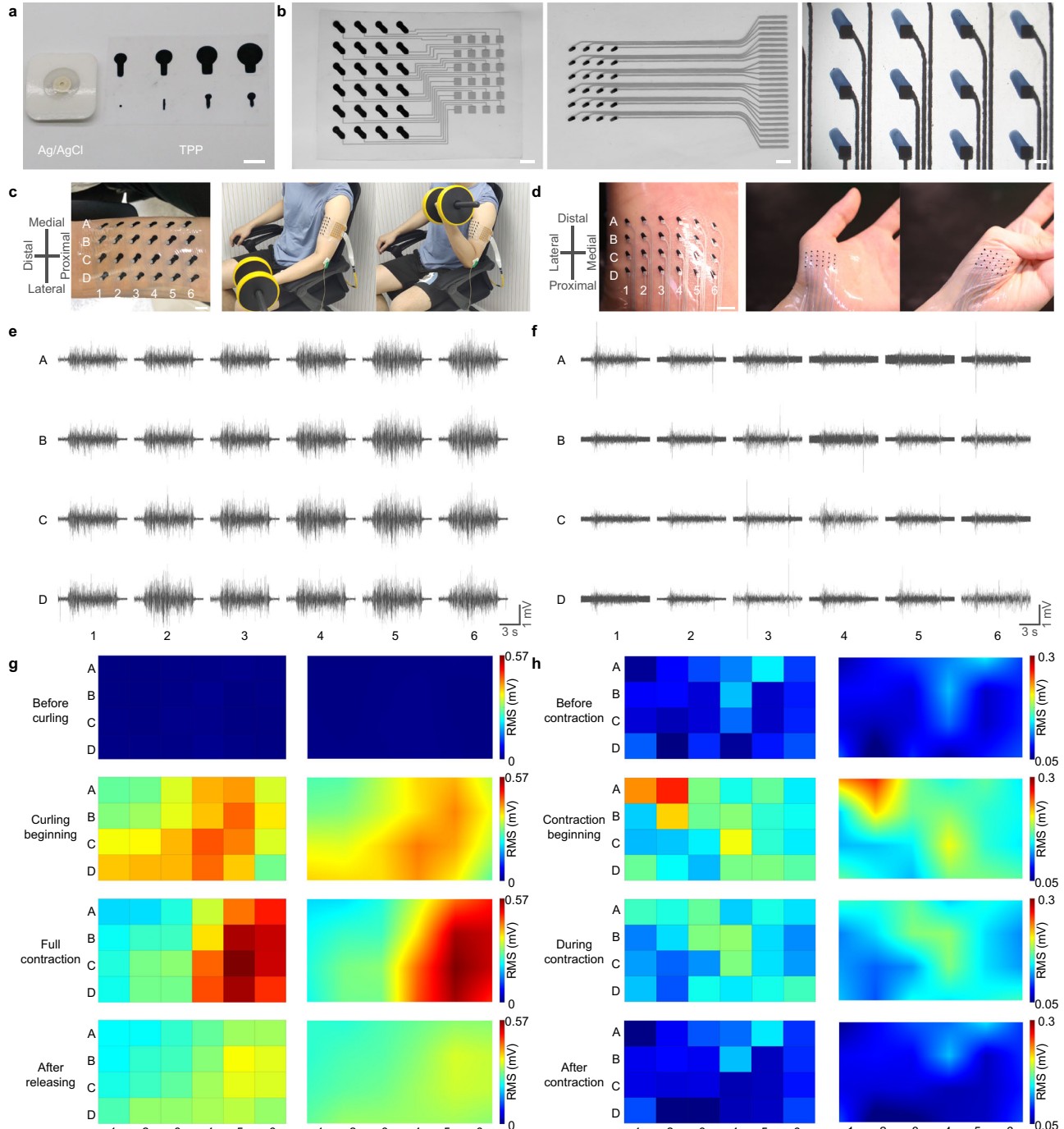

**Fig. 6 | High-density sEMG recording on BB and APB using MEAPs. a** Comparison of surface areas of Ag/AgCl electrodes and TPP electrodes. Scale bar: 1 cm. **b** Photographs of different MEAPs. Scale bar: 1 cm, 5 mm, 1 mm, respectively. **c, d** Configuration of MEAP on BB and APB at rest and during the contraction task of BB, APB, respectively. In the task, left is rest; right is contraction. **e, f** sEMG recorded by each channel of MEAPs on BB and APB, respectively. **g, h** Muscle activity maps of sEMG recorded from BB and APB, showing the process of graded recruitment during contractions. The left column is RMS values from each channel on BB and APB, respectively; the right column is interpolated results of each channel on BB and APB, respectively.

## MEAP for muscle-tendon junction location

Muscle injuries are commonly associated with exercise where the muscle and tendon are excessively stretched. This is currently verified after the injury using magnetic resonance imaging or ultrasound imaging by locating the muscle-tendon junction. The tool's ability to inform the individual about excessive stretching during the task, is significant to reduce the risk of injury, which are not possessed by two imaging techniques above. Current Ag/AgCl electrodes cannot provide tendon position information due to their lack of spatial resolution. In

comparison, high-density MEAP provides a clear contrast in mean frequencies obtained from different channels unravelling the tendon displacement, thus providing a marker to monitor tendon position actively. We verified our MEAP-based findings with ultrasound images of biceps distal tendon in a representative subject while the subject performed the isometric task with load of 5 kg (Fig. 7a, Supplementary Movie 5). The positional difference of muscle-tendon junction was about 3.81 cm between flexion and extension confirmed using ultrasound image (Fig. 7b). Using the 4-column MEAP, junction location was

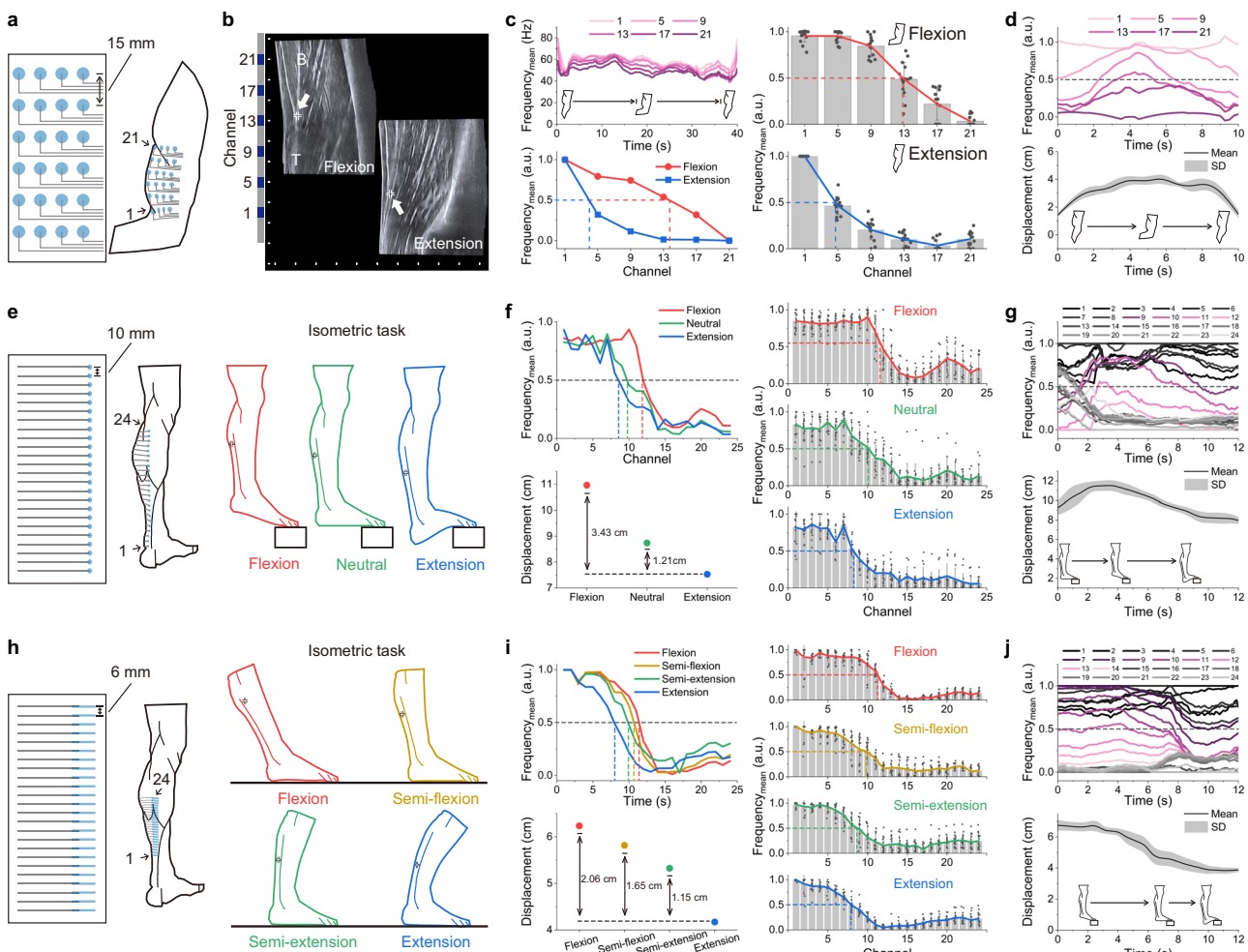

**Fig. 7 | Location of muscle-tendon junction by MEAP. a** Schematic diagram of a MEAP on the biceps. The IED was 15 mm. Channel numbers (1–24) were ordered from left to right and from bottom to top. **b** Ultrasound image of tendon displacement during the isometric task with load of 5 kg and MEAP relative position on the skin. Scale bar: 1 cm. B: biceps brachii; T: distal biceps tendon. **c** Mean frequencies of the EMG signals; left panels show data from the biceps brachii of a representative subject during the isometric task and the normalised mean frequencies for each channel during flexion and extension. Right panels show the normalised mean frequency data in multiple subjects. MEAPs were attached on comparable positions on the biceps brachii muscles of the subjects to obtain junction locations. **d** Normalised mean frequencies of the EMG signals recorded from the biceps brachii of a representative subject during the dynamic task and real-time junction displacement in multiple subjects. **e, h** Schematic diagrams of MEAPs on the gastrocnemius and Achilles tendon, and isometric tasks on a step and on the ground, respectively; the IEDs were 10 mm and 6 mm respectively. Channel numbers (1–24) were ordered from bottom to top. **f, i** Normalised mean frequencies of the EMG signals; left panels show data from the gastrocnemius and Achilles tendon of a representative subject during different isometric tasks and their corresponding displacements. Right panel shows the normalised mean frequency data in multiple subjects. MEAPs were attached on similar locations on the Achilles tendon of subjects to obtain junction locations. **g, j** Normalised mean frequencies of the EMG signals recorded from the gastrocnemius and Achilles tendon of a representative subject during the dynamic task and real-time junction displacement in multiple subjects. All displacements represent the distance between the first channel position and the junction position; all statistical experiments were conducted with the three repeated isometric or dynamic tasks performed by five subjects (n = 15, different MEAPs used); data are presented as mean values ± SD. a.u. arbitrary units.

determined by the column with highest RMS value, which was the first column, because it was closest to the muscle belly horizontally (Supplementary Fig. 21). The mean frequencies of sEMG recorded by the channels in the same column, positioned from muscle to the tendon, differed in ascending order (Fig. 7c). We did normalization based on the first and last channel, which were always located on the tendon and muscle respectively, to explore the change of channels in between. In the isometric task, the channels between showed noticeable difference in the cases of flexion and extension. According to the results of ultrasound image, we found the junction location was always close to the value of 0.5 after normalization. We defined the channel with value of 0.5 is suitable as the muscle-tendon junction position (Supplementary Text 1). This observation was also verified by palpation on the biceps brachii. Recording from five different subjects with a total of 15 MEAPs was carried out, and the mean frequency for each channel is

plotted unravelling distinct trends reflecting flexion and extension, thereby demonstrating the consistent recording capability of the MEAP in identifying junction positions. Similarly, the displacement of the junction was continuously monitored throughout the dynamic task of a biceps curl (Fig. 7d). We also improved calculation to make the influence of skin deformation as low as possible (Supplementary Text 2).

We continued to explore junction location of the Achilles tendon using a 24-row MEAP with IED of 10 mm (Fig. 7e and Supplementary Fig. 22). The subject was asked to isometrically contract the gastrocnemius on a step to stretch the tendon by different lengths. The displacement of the junction was successfully identified in each task and decreased from flexion to extension (Fig. 7f). We also obtained the real-time displacement of Achilles tendon junction accurately when switching from plantarflexion to dorsiflexion even in different subjects

(Fig. 7g). Once the junction movement range was identified, a MEAP with shorter IED of 6 mm was used to further improve the precision of the location (Fig. 7h). Using the smaller IED, tendon stretch was assessed in four lower leg positions on flat ground during the isometric task. The junction location could be identified during four leg positions (Fig. 7i). This indicated MEAP is capable of monitoring junction displacement across different tasks in different scenarios. During the dynamic task on the step, using the MEAP with IED of 6 mm we found more channels were engaged in the junction location compared to that with an IED of 10 mm (Fig. 7j), thus improving the location precision. This furthers the possibility of improved monitoring of tendons and muscles by modifying the resolution of the MEAP as required to personalize the monitoring and diagnostics.

## MEAP for muscle injury prevention

Dumbbell biceps curl is the most common routine used to strengthen biceps brachii, which carries the potential risk for exercisers to have muscle injuries with wrong loading or excessive training (Fig. 8a). In this case, muscle loading, muscle fatigue and tendon displacement can provide comprehensive information for injury prevention and exercise improvement[74] (Fig. 8b). To verify if MEAP can provide such multiplexed information, sEMG was recorded from the biceps of 5 subjects during 5 sessions. Taking subject A as an example, each session included isometric and dynamic tasks, with load from 1 to 5 kg (Fig. 8c, d and Supplementary Fig. 23). The sEMG recorded from the 6 channels in the left column and 4 channels in the top row were used for further detailed analysis due to their highest RMS values compared to other columns or rows (Fig. 8e). The data was statistically verified to confirm that each channel on the MEAP recorded distinct sEMG information from the muscle (Fig. 8f). In addition, it was observed that the activation patterns of the biceps muscle are consistent even among 5 different subjects. It is worth noting that the variability of data increases as the distance between the recording channel and the control channel grows. We speculate that this phenomenon is primarily attributed to variations in muscle length among subjects. Subsequently, for a more comprehensive examination of the sEMG signals captured by MEAP, we proceeded with data analysis focused on the recordings obtained from subject A. In the isometric task, RMS amplitude increased with increasing load (Fig. 8g). For the dynamic task, a similar relation between the RMS amplitude and load for each of the curls was observed. We also noted that the RMS values recorded from channels 13, 17 and 21 which were close to muscle belly were higher than those recorded from 1, 5 and 9 on the distal tendon. At the high load (4 and 5 kg) sessions, RMS values showed obvious increase during last few curls—indicative of fatigue[75]. This result showed that the MEAP can detect muscle loading in both isometric and dynamic scenarios. In addition, MEAP also provided a measure of muscle fatigue, indicated by the decreasing median frequency of sEMG signals[76]. Median frequency changing with time at different channels were showed (Fig. 8h). In isometric task, median frequency change at each channel was linearly fitted to obtain slope values. Results showed that muscle fatigue was seen during the curls starting with 3 kg and increased with 4 and 5 kg. Similarly, during dynamic task, the median frequencies in each curl visibly decreased with loads <3 kg. These results showed that the MEAP can be used to detect the level of fatigue across all channels, as well as muscle loading (Supplementary Fig. 24). We generated a real-time heatmap of RMS, median frequencies and mean frequencies derived from sEMG signals to show muscle condition (Supplementary Movie 6).

For muscle-tendon junction location, the relations between mean frequency and tendon displacement with different loadings were found (Fig. 8i). Notice the displacements increased with higher loads because higher loads cause more contraction of biceps, leading to more tendon elongation than lower loads even though flexion angles were the same in all isometric tasks. For the dynamic tasks, all mean

frequencies of channels in column 1 with load of 5 kg were normalised (Supplementary Fig. 25) and the traces were smoothed to improve the display. The values of the channels increased during flexion, approaching the tendon range and decrease with extension. Using the same determining strategy in isometric task, we can identify the real-time movement of muscle-tendon junction during the dynamic task. Statistical analyses were conducted to show distinction between muscle and tendon (Supplementary Fig. 26 - 28). This can warn exercisers when their tendon displacements are keeping high value in the eccentric contraction, helping injury prevention. To give a comprehensive assessment of the injury possibility, we combined RMS, fatigue slope value and tendon displacement together into consideration to help exercisers protect themselves or do effective training (Fig. 8j). RMS values were normalised based on the results of 1 kg loading to give a universal frame for every exerciser. When a tendon has 8 - 10% strain, it is highly likely to tear[77,78]. As a result, such a high-possible injured circumstance is depicted as red range in the muscle injury index. For other three ranges, each one should be determined specifically by the exerciser under professional instructions. In this context, the loads were classed as safe (<3 kg), effective (3−5 kg) and vulnerable (>5 kg) based on the subject's previous experience. The MEAP successfully provided information about muscle loading, muscle fatigue and tendon displacement of the other subjects, which verified the stable and reliable recording using MEAP across all individuals (Supplementary Figs. 29, 30).

## Discussion

We developed an adhesive, stretchable, biocompatible, and durable sEMG metal-polymer electrode array patch (MEAP) using MPC and TPP as circuit and electrode, respectively, which ensures the excellent flexibility and stretchability of the device (over 200% strain). TPP electrodes during long-term wearing over 10 days on the skin without any inflammation show long-lasting effectiveness and good biocompatibility. The adhesiveness of the electrode (0.58 N/m for peeling force) contributed to high conformability itself on the skin, resulting in better quality and stability of signals than Ag/AgCl electrodes, particularly when recording from skin surfaces with folds and creases. The signal quality is also suitable to detect decrease in median frequency in the sEMG, thus providing the fatigue levels of the muscle. These remarkable properties of TPP electrodes and the patternable fabrication of MPC circuits, help us develop a high-density electrode array patch to record multi-channel sEMG signals. However, commercial hydrogel electrodes face challenges in achieving the same recording sites in the same area as MEAP, and commercial arrays struggle to attain similar conformability as MEAP. This difficulty is simply solved by using patternable liquid metal circuitry. We must underline this originality once again. These MEAPs offer precise information about muscles and monitor muscle fatigue, providing a measure of muscle performance during activities like sports and training. For example, erector spinae injury is common in weightlifting, impeding regular training. Common sEMG electrodes on the back cannot stay in place owing to movement of the skin underneath, stretching and relaxing, but the MEAP makes a conformal attachment instead, which would hold promise to help monitor the muscle activity and prevent injuries. This tool can be used in a lot of sports and training environments, particularly useful for those requiring explosive strength such as sprinting and bodybuilding. More importantly, conveniently locating the muscle-tendon junction using the mean frequency changes is only feasible because of the high-density sEMG electrode array. Real-time monitoring of tendon displacement based on MEAP recordings, to maintain tendon stretch within appropriate physiological range, will prevent injuries to exercisers. Depending on information about loading, fatigue and tendon displacement obtained by MEAP, personal data analysis can help exercisers work more safely and efficiently. In addition, sizes and designs of MEAP can be changed easily for different

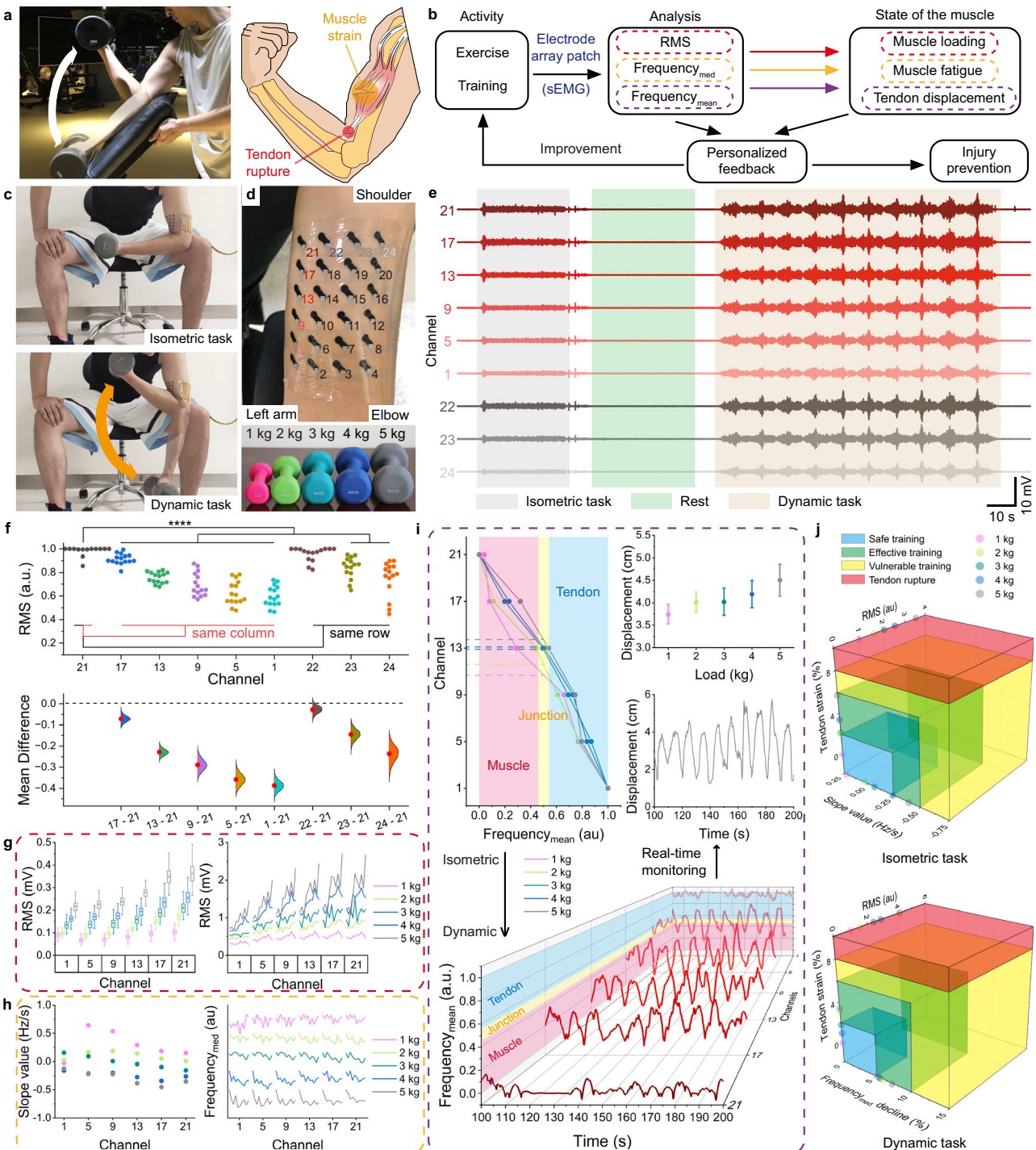

**Fig. 8 | High-density sEMG recording by MEAP for muscle injury prevention. a** A photograph of a biceps curl and a risk schematic. **b** Diagram of the injury prevention system components. **c** Photographs of a session including isometric and dynamic tasks. **d** Photographs of channel sites of MEAP and different loads. **e** sEMG signals in 5 kg load session. **f** Statistical analysis of sEMG signals recorded by MEAP on different subjects. The Gardner-Altman plot illustrates the RMS values of sEMG signals captured by MEAPs ($n = 15$, 3 repeated isometric tasks performed by five subjects, different MEAPs used; mean presented as red dots, 0.95 confidence interval presented as red line, and Kernel Smooth used for distributions). The RMS values are normalised to their respective maximum values, and channel 21 (control) is compared with others. Significance was determined by one-side $t$ test (*$P < 0.05$; **$P < 0.01$; ***$P < 0.001$; ****$P < 0.0001$). **g** RMS values of sEMG signals in the isometric task (left) and in the dynamic task (right) across selected channels ($n = 54$ RMS values per channel). The box plots show the mean (centre square), median

(centre line), the 25th to 75th percentiles (box) and the smallest and largest value that is ≤1.5 times the interquartile range (the limits of the lower and upper whiskers, respectively). **h** Median frequency details of sEMG signals recorded by MEAPs. Linear fit slopes for median frequencies in the isometric task (left) and changes of median frequencies in the dynamic task (right) across select channels are shown. **i** Top left, plot of normalised value of mean frequencies among six channels in the isometric task; top right, plot of tendon displacements in the isometric task ($n = 3$ repetitive tasks for each measurement); bottom, plot of normalised values of mean frequencies among six channels in the dynamic task; middle right, plot of real-time monitoring of the tendon displacement in the dynamic task. **j** A visual representation of the potential for muscle injury index, generated based on the assessments made using the MEAP. The assessment is presented as a unified model using the measures obtained from (**g–i**).

muscles or other electrophysiological recording such as EEG and ECG. Thanks to high spatial resolution of recording and long-lasting working time, this tool can also play a role in human-machine interface, e.g., prosthetics or virtual reality. We also aim to replace invasive needle electrodes in clinical applications with MEAP to provide patients with improved comfort. However, current MEAPs have difficulty recognizing single motor unit signals from sEMG recording because the lack of intelligent back-end algorithms causes low single motor unit selection efficiency and accuracy. In this case, MEAP can only provide limited help to clinical diagnosis. Another drawback of current MEAPs is they are using cable connection, which limits the application scenarios and simplicity. To address these issues, we are endeavouring to combine MEAP with intelligent algorithms and wireless modules to make whole devices more useful in clinic scenario and more portable in daily life. We believe in the future, MEAP has enormous potential to be commercialized because of its low cost and simple fabrication, thus providing a platform for disease diagnosis, daily rehabilitation management and scientific exercise.

## Methods

### Materials

PEDOT:PSS (Clevios PH 1000) aqueous solution was from Heraeus Co. The concentration of PEDOT:PSS is 1.0 ~ 1.3 wt%. EgaIn (Gallium Indium eutectic, 99.99%), Tannic acid (Sigma-Aldrich T0200-250G) and dimethyl sulfoxide (DMSO, ACS, ≥99.9%, Aladdin) were from Sigma-Aldrich. Polyvinyl alcohol 1799 (PVA) and n-Decyl alcohol (98%) were from Macklin Inc. Polydimethylsiloxane (PDMS, Sylgard184) were from Dow Corning Co.

### The fabrication of PEDOT-PVA and TPP electrode

PEDOT:PSS solution was filtered by a PTFE syringe driven filter (0.45 µm, Jet Bio-Filtration Co., Ltd.) before use. The PEDOT-PVA solutions were made by each adding 0, 0.01927, 0.0288, 0.0575, 0.115, 0.1725 g PVA into 5 ml PEDOT:PSS solutions with 250 µl DMSO, whose weight fractions of PVA were 0, 25, 33.3, 50, 66.7 and 75 wt%, respectively. The solutions were stirred at 80 °C for 4 h to be prepared. For TPP electrodes, PEDOT-PVA solution was made by controlling the ratio between PVA and PEDOT as 2:1. A three-fold dilution was made to the PEDOT-PVA solution. The TPP solutions were made by each adding 0, 0.024, 0.048, 0.096, 0.144, 0.192, 0.24, 0.288, 0.336, 0.384 g TA into 3 ml diluted PEDOT-PVA solution with 150 µl DMSO totally included, whose weight fractions of TA were 0, 0.8, 1.6, 3.2, 4.8, 6.4, 8, 9.6, 11.2, 12.8 wt%, respectively. The TPP solutions were stirred for 5 min to be prepared.

### The fabrication of MEAP

3 g EgaIn and 1 mL n-Decyl alcohol were added into a 5 mL centrifuge tube and sonicated by a sonicator (Branson SFX 550, Emerson Electric Co.). For screen printing, the EgaIn was sonicated for 1 min with the power of 300 W. Printing equipment and masks were bought online to print required circuit patterns on polyethylene terephthalate (PET) substrate. The PDMS (base: curing agent ratio of 10:1) was spin coated on the PET substrate with 1000 rpm, 30 s. The PDMS films were peeled off from PET films after curing in 80 °C for 20 min. An electrode mask made by silicone film (0.2 mm, Shengyuwujin, China) was attached to the circuit side of PDMS film. After the plasma treatment, the mask was removed, and 66.7% TPP solution was introduced on the electrode location. Because only the areas that were to become the electrodes were plasma-treated to become hydrophilic, TPP solution would only cover these areas to form the electrodes according to the shape of the design. After heating in 80 °C for 10 min, the flexible electrode film was made. The adhesive PPA were spin-coated (2000 rpm, 30 s and 3000 rpm, 60 s) on PDMS for encapsulation of the liquid metal circuits to prevent leakage. Note that electrode sites and connection pads were protected by silicone films during the encapsulation to allow the electrodes and connection pads to be exposed. After curing in 80 °C for 20 min, the soft electrodes were completed. The thickness of MEAPs was lower than 100 µm, typically with the substrate of 65 µm, the encapsulation layer of 25 µm and the TPP electrode of 30 µm. We can also directly use medical infusion stickers as encapsulation layer to accelerate fabrication. For comparison between commercial array, the diameters and inter-distances of TPP electrodes were 4 mm and 8 mm. For BB recording, the diameters and inter-distances of TPP electrodes were 5 mm and 15 mm, respectively. For APB recording, the diameters and inter-distances of TPP electrodes were 1 mm and 5 mm, respectively. For the Achilles tendon location, the MEAP #1 with IED of 10 mm had electrode surface area of 20 mm$^2$ and the MEAP #2 with IED of 6 mm had electrode surface area of 30 mm$^2$. The front-end connectors were made by polyimide flexible printed circuit (FPC). Front-end connectors were designed and fabricated (EasyEDA, China) for connecting specific MEAPs. The FPC and MEAP were hot-pressed together with force of 50 N and temperature of 140 °C for 30 s by a hot-pressing machine (G311, Freamc, China). With a customized back-end connector, every channel of MEAP can be independently connected to EMG recording system.

### Materials characterization

The abrasion test was performed by an RCA abrasion wear tester (Norman Tool Inc., USA). A paper ribbon was imposing a consistent rubbing force of 1.715 N on the electrode. The resistance of PEDOT-PVA strip was monitored during the abrasion test. When the resistance had more than a hundredfold change, the PEDOT-PVA was regarded as damaged by the paper ribbon. All electrode strips of 50 × 5 mm kept being rubbed until they cracked. The water resistance test was conducted in a water bath of a sonicator (skymen JP-040S*, Skymen Cleaning Equipment Shenzhen Co.,Ltd). For viscosity test, different PEDOT-PVA solutions were tested using HR 10 Discovery Hybrid Rheometer (TA Instruments, USA) at 25 °C, 1% of the strain and 1 Hz frequency. The topography of PEDOT-PVA film was characterized by SEM (ZEISS Merlin, Germany) and AFM (MFP-3D Stand Alone, Asylum Research, USA). For the cross-sectional SEM images, TPP films were freeze-dried overnight followed by quenching in liquid nitrogen. The tensile testing was performed by a universal testing system (Instron 68TM-5, USA), size of PEDOT-PVA and TPP films was 30 mm long and 10 mm wide. The stroke speed of the measurement was 0.5 mm min$^{-1}$. The peeling force measurement was conducted on the forearm. Before the measurement, the forearm skin was cleaned with a mixed solution of deionized water and ethanol. The TPP films were made 20 mm wide and 20 mm long. During the test, the stroke speed was kept at 15 mm s$^{-1}$.

### Impedance measurement

Electrode-skin impedance measurement was performed with Multi Autolab/M204 potentiostat (Metrohm, Switzerland), with 10 frequencies per decade within a frequency range from 10 Hz to 10 kHz considering that most of the sEMG responses are in the 20–500 Hz range. A sinusoid stimulation voltage, with RMS amplitude of 10 mV and no direct current offset, was applied to obtain the impedance curve. For both PEDOT-PVA and TPP electrodes, two working electrodes were attached over flexor carpi ulnaris (FCU) muscle with an IED of 20 mm, on the internal part of the forearm, and reference electrode was attached on the elbow, on the external part of the forearm. To compare, Ag/AgCl electrodes (Foam Monitoring Electrode 2228, 3 M, USA) were used to achieve impedance results using the same process. Both types of electrodes had the surface contact area of 20 mm$^2$.

### Biocompatibility of TPP film and cell viability

For the long-term test, the subject was asked to behave the same as daily life, including work, exercise and taking shower regularly. The

sEMG recording was done every noon to monitor performances of electrodes. For cell viability tests, PPA, LM, PDMS, PEDOT-PVA and TPP films were sterilized under UV light overnight. Especially, the TPP film was treated with desterilized $FeCl_3$ solution (10 mg/mL) for 48 h to stabilize the film on a slide glass. HUVECs (ATCC, USA) were seeded at $10^5 mL^{-1}$ on the film and cultured in Dulbecco's modified eagle medium (DMEM, Gibco, USA) supplemented with 10% foetal bovine serum (5% $CO_2$, 37 °C). After 5 days, HUVECs were stained with live/dead kit: Calcein-AM (green, live cells, Solarbio, China), PI (red, dead cells, Solarbio, China), and DAPI (blue, nucleus, Sigma, USA). The confocal microscope (ZEISS LSM 980, Germany) was used to take fluorescent images.

### sEMG signal recording

The electrodes were connected to g.Hiamp multi-channel amplifier (G.tec, Austria) for sEMG signal acquisition with a sampling frequency of 1200 Hz and an analogue notch filter at 48 - 52 Hz. The skin was cleaned with alcohol before recording. Foam Monitoring 2228 electrodes were used for long-term test, conformability test on the forehead, and Red Dot 2223 (USA, 3 M) electrodes were used for fatigue comparison tests. For fatigue comparison tests, electrodes were attached on the most prominent bulge of the muscle belly of FCU, and different types of electrodes were attached exactly on the same position. 2228 and 2223 electrodes have the surface contact area of 2 and 2.4 $cm^2$. The TPP electrodes in each comparison test have the same surface contact area with 2228 or 2223 electrodes correspondingly. All interelectrode distance for bipolar recording is 20 mm. Bipolar recording was used for single-channel TPP electrode and monopolar recording was used for MEAP to obtain sEMG signals, unless otherwise noted. During the biceps brachii muscle recordings, a comparative test between PEDOT-PVA and TPP electrodes was performed. The subject was instructed to keep the curl speed between flexion and extension at 4.5 rad/s. Five contractions of each electrode were tested for RMS alterations. For long-term test, three contractions were recorded each day to provide statistical RMS and SNR values. The commercial array has 64 channels Ag/AgCl electrodes on a polyimide substrate with thickness of 100 μm (Neuracle, China). The conductive gel g.GAMMAgel (G.tec, Austria) was used between commercial array and the skin. For all sEMG recording, no skin treatment was used, including shaving, rubbing or cleaning of the skin. All sEMG signals were filtered with a 20 Hz Butterworth infinite impulse response high-pass filter, which is recommended for general use[79].

### Muscle contraction task for muscle activity maps

For BB recording, the subject performed a biceps curl (concentric and eccentric activity) starting from a full extension of the elbow to a full flexion, with a 4 kg dumbbell while sEMG signals were recorded from the muscle with all 24 sites. The task performed lasted for 10 s, and then the subject released the dumbbell to the starting position. For APB recording, the task was moving thumb to the proximal end of little finger and moved back. The RMS values of the recorded sEMG were calculated for timesteps of 0.25 s and 0.083 s for BB and APB, respectively. Muscle activity maps were generated to help visualize the change in activity during the task.

### Muscle contraction task for muscle-tendon junction location

Five male subjects, with average age of 23.4 ± 1.9 (mean ± SD), were recruited for the task. Ultrasound image was achieved by Mindray L14-6NE probe (Mindray DC-40 Ultrasound System, Lysis Healthcare GmbH, Austria). The isometric task for biceps brachii was holding the 5 kg dumbbell for 30 s, with forearm paralleled to the ground and the arm with a right angle to the forearm. The dynamic task was making a biceps curl with the 5 kg dumbbell including bending and extending.

The isometric task for the Achilles tendon recorded by MEAP #1 was maintaining three lower leg positions, with the gastrocnemius naturally contracting. The isometric task for the Achilles tendon recorded by MEAP #2 was maintaining four lower leg positions, with the gastrocnemius making voluntary contraction. The dynamic task for both MEAPs were the same that the subject did plantarflexion to dorsiflexion with their tiptoe on the step.

### Muscle contraction task for muscle injury prevention

The same five subjects mentioned earlier used their non-dominant hand to perform one-arm concentration curls to record sEMG signal generated from biceps brachii muscle by MEAP. The MEAP had 4 columns and 6 rows; column 1–4 and row 1–6 were from left to right and bottom to top, respectively. The isometric task started at 10 s and kept for 30 s; after 1 min rest, 10 curls in dynamic task were performed to complete one session. The subjects were asked to control curl speed around 4.5 rad/s for 5 total sessions by the order of 1–5 kg loadings. For the location of muscle-tendon junction, when $0.45 \leq frequency_{mean} \leq 0.55$, the channel was considered on muscle-tendon junction. The tendon strain was calculated based on the average distal biceps tendon length[78]. All human experiments were conducted with the approval from the Medical Ethics Committee of Southern University of Science and Technology (approval no. 2021SYG049). Informed consent was obtained from all participants involved in this entire study.

### Data collection and analysis

Impedances were collected and analysed by NOVA (Metrohm Autolab, V2.1.4). sEMG data were collected by g.Recorder (g.tec, V5.16.01) and analysed by OT BioLab+ (V1.5.9) and Origin Pro 2021. All RMS, median frequency, and mean frequency values of the recorded sEMG signals were computed for time steps of 0.125 s, unless specified otherwise. For dynamic tasks in muscle-tendon junction location section, the mean frequency values of sEMG data were initially smoothed using a Savitzky−Golay filter (with a frame length of 21 and an order of 1) in Matlab R2020a; for real-time monitoring of dynamic tasks in the same section, each set of values were determined first and then the means were generated and plotted as mean ± SD. In the section of injury prevention, the Gardner-Altman plot was generated with a confidence level of 0.95 and a total of 5000 bootstrap samples.

### Statistics and reproducibility

Data are presented as mean values ± SD, unless otherwise noted in the figure caption. Significance was defined as *$P < 0.05$; **$P < 0.01$; ***$P < 0.001$; ****$P < 0.0001$. Statistical analysis was performed using Origin Pro 2021.

### Reporting summary

Further information on research design is available in the Nature Portfolio Reporting Summary linked to this article.

## Data availability

The sEMG data generated in this study have been deposited in the Science Data Bank database. This is under restricted access due to commercial sensitivity as it is awaiting patent approval. Access for academic use can be obtained upon request from the authors.

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

## Acknowledgements

We thank the National Key R&D Program of China (2021YFF1200800, 2021YFF1200100, 2022YFB3804700, and 2018YFA0902600), the National Natural Science Foundation of China (22234004), Shenzhen Science and Technology Program (JCYJ20200109141231365 and KQTD 20190929172743294), Shenzhen Key Laboratory of Smart Healthcare Engineering (ZDSYS20200811144003009), Guangdong Innovative and Entrepreneurial Research Team Program (2019ZT08Y191), Guangdong Provincial Key Laboratory of Advanced Biomaterials (2022B1212010003), Tencent Foundation through the XPLORER PRIZE, Guangdong Major Talent Introduction Project (2019CX01Y196). We thank Dr. Christopher Russell and Dr. Rafael De Castro Aguiar for the constructive suggestions. We thank Zhuowei Feng for the professional photography. We also acknowledge the assistance of SUSTech Core Research Facilities.

## Author contributions

X.J., S.C. and S.Y. conceived the idea. S.Y. designed experiments, fabricated devices and wrote the paper. S.Y., J.C., J.S., J.Q., L.Z. and L.D. characterized the MEAP. C.H. and Q.R. completed cell viability tests. S.Y., J.C., J.S., L.H., C.L. and M.Z. conducted sEMG recording. S.Y. and S.C. analyzed the data. S.C. and X.J. supervised the study, interpreted the results, and revised the paper. All the authors took part in the discussion and writing.

## Competing interests

X.J. and S.Y. are coinventors on international patent application no. PCT/CN2022/088644, "Fabrication and application of a stretchable sEMG electrode array". The remaining authors declare no competing interests.
