## [Peer Review File · Nature Communications]

Reviewers' comments:

Reviewer #1 (Remarks to the Author):

Yang et al. reports the fabrication of stretchable surface electromyography electrode array patch enabled by the combination of an adhesive dry electrode array and metal-polymer electrode array patch. The resulting electrode array shows long-term application capability with good durability and biocompatibility. The authors demonstrated the electrode array by the application of tendon location and muscle injury prevention. This study is well organized and the performance of the electrode array are well characterized. The results are interesting, which should be impacts on the design of dry electrodes and the development of surface electromyography. Therefore, I would like recommend the publication of this work by minor revision.

1)The mechanical properties of the electrodes are important for the practical application as dry electrodes for skin adhesion. It is suggested to provides some details characterization of the electrodes, such as modulus, stretchability, and reversibility.

2)The resolution of figures needs to be improved.

3)The authors claimed that “As a result, how to fabricate a conformal, adhesive and robust dry electrode becomes an issue to address” in the introduction. Actually, to address this issue, some recent work (cited as Ref. 3) based on the supramolecular solvent (β -cyclodextrin and citric acid), PVA and PEDOT:PSS shows soft and conformal adhesive properties as a dry electrode for the monitoring of physiological electric signals as well as flexible electronic devices. Therefore, it is suggested to provide some in depth discussion for this issue in the introduction part.

Reviewer #2 (Remarks to the Author):

This manuscript reports a stretchable surface electromyography electrode array patch, which was made by integrating liquid metal based stretchable circuit with PEDOT:PSS based soft dry electrodes. The authors first carefully checked the performance of the electrode and then showed its potential applications in predicting muscle-tendon junction location and muscle injury prevention. The research topic is interesting and the work presents some impressive results. Here are some comments which should be addressed before its consideration for publication:

1. The PPT dry electrode, as a highlight of the paper, is critically important to the recording of sEMG. However, it seems that the same material with similar recipe has already been reported and used for epidermal biopotential measurement in the literature [1]. What's the novelty and challenge of the PPT dry electrode in this paper?
2. For Figure 6, the sEMG signal pattern is strongly and frequently influenced by the innervation zone [2, 3], in this case, neither the RMS or the mean frequency could be monotonous. How to predict the muscle-tendon junction location?
3. Why is value of 0.5 suitable as the muscle-tendon junction position, please explain or add related reference.
4. The inter-electrode distance could be largely changed during the isometric task from flexion and extension due to the deformation of the sEMG array patch adhered on muscle, how to define its effect?
5. For Figure 4g, we agree that decreasing median frequencies indicated fatigue of the muscle. However, the authors claimed the decreasing value of slopes of median frequencies indicated the fact that muscle became more fatigued. If so, the slopes of median frequencies during each task (30 s) should not be a constant since the muscle is getting fatigue as the time goes. Please have a check and related reference are needed.
6. As shown in Figure 3d, TPP electrodes can work superbly with SNR level above 20 dB for almost 5 days, and then get worse. Please give the reason that decreases the SNR of PPT electrodes for long-term measurement up to 5 days.
7. How about the adhesion, skin-electrode impedance and SNR of the sEMG electrode array patch on the skin when after, e.g., 200 times, movements (compress or stretch)?
8. Could be the sEMG electrode array patch be used repetitively?
9. Please specific the thickness of each layer of the sEMG electrode array patch.
10. Bipolar recording was used for single-channel TPP electrode (Figure 4e) and unipolar recording was used for MEAP, why and what's the different?

Reference

- [1]. Cao, J. et al. Stretchable and Self-Adhesive PEDOT:PSS Blend with High Sweat Tolerance as Conformal Biopotential Dry Electrodes. *ACS Appl. Mater. Interfaces* 14, 39159–39171 (2022).
- [2]. Farina D, Madeleine P, Graven-Nielsen T, et al. Standardising surface electromyogram recordings for assessment of activity and fatigue in the human upper trapezius muscle[J]. *European journal of applied physiology*, 2002, 86(6): 469-478.
- [3]. Beretta Piccoli M, Rainoldi A, Heitz C, et al. Innervation zone locations in 43 superficial muscles: toward a standardization of electrode positioning[J]. *Muscle & nerve*, 2014, 49(3): 413-421.

Reviewer #3 (Remarks to the Author):

The manuscript describes a surface electromyography electrode, which is novel, as the authors state, because it is characterised by the fact that it adheres adhesively to the skin surface, is stretchable and forms an array (see abstract). This claim by itself shows that the authors seem to be unfamiliar with the state-of-the-art in surface electromyography (sEMG). Adhesive sEMG electrodes that adhere independently to the surface of the skin have been available for several years. The manuscript does not comment on this, nor does it compare the supposedly so good new electrode with it. Instead, an unspecified Ag/AgCl electrode is used for comparison, which, as can be seen from the figures, does not correspond to the standard for sEMG electrodes. It is therefore doubtful to what the electrode introduced in the manuscript is compared to and how meaningful this comparison is. Electrode arrays that adhere to the skin surface for long periods of time have also been described since the 1990s and are now commercially available. There is no reference to this in the manuscript either, nor is the introduced electrode compared to them.

This leaves the property of stretchability, which according to the authors should improve the quality of sEMG signals. The advantages and disadvantages of stretchable electrode arrays have been debated among sEMG experts for many years. The problem is, that the interelectrode distance changes when the array is stretched. This affects the frequency spectrum of the sEMG signal in the case of a bipolar lead. Investigations in the frequency domain, as suggested by the authors for fatigue detection, are therefore not valid for non-isometric contractions, as it is not possible to exclude beyond doubt that a measured change in the frequency domain is not due to a change in the electrode distance. Stretchable electrode arrays are therefore fundamentally unsuitable for such applications.

This brings me to another problem concerning the manuscript. The manuscript is full of claims - often in the superlative - about signal quality and possible applications of the described electrode, which are not statistically proven. They seem to be the purely subjective perceptions of the authors. This becomes particularly clear in Fig. 2 k, in which a signal with a motion artefact, which occur from time to time but not regularly, was compared with the signal detected with the introduced electrode. The manuscript does not describe whether and if so how repeat measurements were carried out and how these were statistically evaluated to substantiate the statements made.

Fig. 4 a and e shows another problem that arises when characterising the quality of the novel electrodes. The electrodes of the devices used for comparison are not located in the same position as the novel electrodes. Rather, the comparison signals are derived at less favourable positions, which has a negative influence on the signal amplitude, the SNR and the frequency spectrum. An objective comparison between the two devices is not possible under these conditions.

Finally, a comment on electrode arrays. The use of electrode arrays has been known for a long time under the pseudonym High Density sEMG (HDsEMG) and is widely used in different research questions. The method is called sEMG imaging and the "heat maps" shown in Fig. 5 g and h are called "muscle activity maps" in the literature. The fact that HDsEMG is suitable for localising anatomical structures such as the neuromuscular junction or tendon insertion has been known since the 1990s and has been studied in a number of different investigations. In connection with fatigue and pain, a change in the spatial distribution of the activity of the muscle has already been demonstrated, as well as a change in

the spectrum of the signal. This fundamental work is not mentioned anywhere in the manuscript. Rather, the impression is given that such investigations are only made possible by the new type of electrode.

1 **Response to reviewers for the manuscript (NCOMMS-22-46103A-Z)**

**Reviewer #1 (Remarks to the Author):**

*Yang et al. reports the fabrication of stretchable surface electromyography electrode*
*array patch enabled by the combination of an adhesive dry electrode array and*
*metal-polymer electrode array patch. The resulting electrode array shows long-term*
*application capability with good durability and biocompatibility. The authors*
*demonstrated the electrode array by the application of tendon location and muscle*
*injury prevention. This study is well organized and the performance of the electrode*
*array are well characterized. The results are interesting, which should be impacts on*
*the design of dry electrodes and the development of surface electromyography.*
*Therefore, I would like recommend the publication of this work by minor revision.*

**Our response:** We appreciate the reviewer taking the time to carefully read our
manuscript and provide such excellent feedback.

*1)The mechanical properties of the electrodes are important for the practical*
*application as dry electrodes for skin adhesion. It is suggested to provides some*
*details characterization of the electrodes, such as modulus, stretchability, and*
*reversibility.*

**Our response:** We appreciate the reviewer pointing out the lack of mechanical

characterizations. Following the reviewer's advice, we conducted a few studies,
including tensile testing and repeated stretch measures. We agree that these findings
are significant, so we include them in Fig. 2.

Meanwhile, we transfer the original Fig. 2j to Supplementary Fig. 7.

Our modifications on Page 8:

Lower Young's modulus gives a better compliance and stretchability to the film,
which is are vital to the conformal adhesion between electrodes and skin⁵⁸. This was
also proved by the results of tensile and peeling tests of TPP films with increasing
concentration of TA (Fig. 2h, i). Such observations helped us use the final weight
concentration of TA at 8%, which makes the film soft and adhesive but not easy to
tear. This TPP film shows elongation at break of 188%, Young's modulus of 644 kPa
and adhesive forces of 0.58 N/cm on the skin. Once the concentration of the
constituents of the TPP solution was determined, each constituent's indispensability
was verified by the changes in conductivity, stretchability, and adhesiveness of the
electrode (Supplementary Fig. 7). Meanwhile, such TPP film showed good
repeatability after being stretched to a strain of 20% for 1000 cycles (Fig. 2j).

...Scale bar: 4 μm ; inset: 1 μm .

52 h Tensile stress–strain curves, strain and Young’s modulus of Strain of TPP films.

i Peeling force of TPP films on the skin.

j Real-time monitoring of the TPP film by stretching the film from a strain of 0 to
20% for about 500 cycles.

k EMG signals recorded by PEDOT-PVA and TPP electrodes. Electrode...

See Page 32: ‘The tensile testing was performed by a universal testing system (Instron

68TM-5, USA), size of PEDOT-PVA and TPP films was 30 mm long and 10 mm

wide. The stroke speed of the measurement was 0.5 mm min⁻¹.’

*2)The resolution of figures needs to be improved.*

Our response: We thank the reviewer for the requested change in resolution. We

provide updated PDF document with higher resolution. We have Tag Image File

Format for each figure if this manuscript gets published.

3)The authors claimed that “As a result, how to fabricate a conformal, adhesive and
robust dry electrode becomes an issue to address” in the introduction. Actually, to
address this issue, some recent work (cited as Ref. 3) based on the supramolecular
solvent (β -cyclodextrin and citric acid), PVA and PEDOT:PSS shows soft and
conformal adhesive properties as a dry electrode for the monitoring of physiological
electric signals as well as flexible electronic devices. Therefore, it is suggested to
provide some in depth discussion for this issue in the introduction part.

**Our response:** We value the reviewer's careful and thoughtful feedback on our
manuscript. This statement prompted us to consider explaining material choices for
flexible electronics. Conductivity is the basis, which allows materials to be classified
into four types: metal, carbon materials, hydrogels, and conductive polymers. Normal
metal (except liquid metal) cannot be stretched unless modified to special structure,
and it is extremely difficult to impart adhesiveness on the metal itself; carbon
materials, such as carbon nanotubes and graphene, have a high Young's modulus,
making them unsuitable for bioelectronics (Matter (2022) 5, 1104-1136). In
comparison, employing hydrogel and conductive polymers that can be fine-tuned
become advantageous tactics since researchers may provide them specific
functionalities depending on the application scenarios. Unfortunately, it is quite
difficult to create a material that is ideal in every way. Sometimes improving one
property of a material implies sacrificing another. For example, hydrogel has a lot of
water but dehydrates quickly; when the conductive polymer is more stretchable, it
becomes less conductive. This is also mentioned in Ref. 3 (Nature Communications
(2022) 13:358), which states: '*Taking into account the compromise in mechanical
flexibility, conductivity, and interface adhesion (in subsequent discussions), SACPs
with PEDOT:PSS mass ratio of 3.6% presented suitable mechanical property
(modulus of 401.9 kPa) and conductivity (3.79 S/cm) meet the requirements of
bioelectrode.*'

We also need to point out that Ref. 3 is an excellent contribution to the field of dry
electrode, but our work has a slightly different focus to Ref. 3. The main point of Ref.
3 is to show the potential of SACPs for future bioelectronic devices, for example, that
making a better tool to visualize EMG. The focus of our work is to provide an array
that can monitor EMG over time to provide detailed information from different

features of muscles. Additionally, the applications of SACPs in Fig. 5 and 6 of Ref. 3
didn't mention an array, which increased our appreciation for the array design of our
MEAP because commercial gel electrodes cannot accomplish the same recording sites
in the same area as MEAP. This difficulty is simply solved by using patternable liquid
metal circuitry. We must underline this originality once again.

Considering the above, we add the paragraph on Page 4: 'As a result, how to fabricate
a conformal, adhesive and robust dry electrode becomes an issue to address. Dry
electrodes force the material to be classified into three types: metal, carbon materials,
and conductive polymers. Conventional metal and carbon materials have exceedingly
high Young's modulus, which must be fabricated into micro-/nano-structures using
complicated procedures for flexible bioelectronics. In addition, employing conductive
polymers that can be variably tuned becomes a more advantageous method since
researchers may provide them specific functionalities depending on the application
scenarios.'

We also add results of conductivity of TPP films with different TA loadings in
Supplementary Fig. 9:

See page 8: 'The conductivity and electrode-skin impedance of TPP film were also
examined (Supplementary Fig. 9, 10).'

We thank the reviewer once more, for the thoughtful assessment and comments. We
appreciated the chance to respond to the reviewer's suggestions in this revised

manuscript because we considered their advice to be really insightful.

**Reviewer #2 (Remarks to the Author):**

*This manuscript reports a stretchable surface electromyography electrode array*
*patch, which was made by integrating liquid metal based stretchable circuit with*
*PEDOT:PSS based soft dry electrodes. The authors first carefully checked the*
*performance of the electrode and then showed its potential applications in predicting*
*muscle-tendon junction location and muscle injury prevention. The research topic is*
*interesting and the work presents some impressive results. Here are some comments*
*which should be addressed before its consideration for publication:*

**Our response:** We value the reviewer's time spent reading our manuscript thoroughly
and providing generally encouraging feedback. We are quite appreciative that the
reviewer found the material to be innovative and recognized its potential for use in
sports health and injury prevention.

*1. The PPT dry electrode, as a highlight of the paper, is critically important to the*
*recording of sEMG. However, it seems that the same material with similar recipe has*
*already been reported and used for epidermal biopotential measurement in the*
*literature [1]. What's the novelty and challenge of the PPT dry electrode in this*
*paper?*

**Our response:** We appreciate the reviewer's comment. Prior to submission we did
find this article (Ref. 1), but it is important that the differences are carefully noted.
The composition of ours are distinctly different from those mentioned in the article,
resulting in our electrodes being very different in corresponding properties. A
comparison table is presented below to show the difference in properties clearly and
this was mentioned in Supplementary Table 1 in the original manuscript.

	Material s for substrates	Multi- channel ?	Young's modulus	Strain	The adhesiveness of electrode (N/cm)	The smallest area of the electrode (mm ²)	Electrode-skin impedance at 100 Hz (KΩ*cm ²)	Long-term test (Hour)	RMS of Noise (μV)	Signal-to- noise ratio (dB)
Ref.1	N/A	No	18.3 MPa	54%	0.28	16	100	N/A	11.8	34.96
This work	PDMS	Yes	645 kPa	188%	0.58	0.8	80	120	1.0	42.3±0.7

Our dry electrode has a far lower Young's modulus than theirs, which makes it softer,
stickier, and more stretchable. The recorded EMG signal demonstrates how each of
these elements helps electrodes adhere to the skin more effectively. Our electrodes'
baseline noise decreased by a factor of ~10 than theirs, which ultimately results in a
greater SNR. In order to investigate muscle loads, exhaustion, and tendon
displacements, high-quality recording is essential. Our work has more potential for
numerous applications due to its long-term stability (a lifetime of 5 days). More
importantly, most of these electrodes—including Ref. 1—only perform the same
function as Ag/AgCl hydrogel electrodes. The more literature we study, the more we
believe that our liquid metal circuits in the patch is significantly distinct and superior.
The liquid metal circuits enable multiple electrodes to record simultaneously over
extended periods of time. We can create an array patch using this stretchable circuit,
which further allows us to map muscle activity and locate muscle-tendon junctions.
The fact that these measurements cannot be performed using Ag/AgCl hydrogel
electrodes must be emphasized. In this paper, we think the dry electrode array is much
more unique than the dry electrode itself.

To better clarify our novelty, we added following sentence in Discussion.

See page 29: **‘However, commercial hydrogel electrodes cannot accomplish the same**
**recording sites in the same area as MEAP. This difficulty is simply solved by using**
**patternable liquid metal circuitry. We must underline this originality once again.’**

*2. For Figure 6, the sEMG signal pattern is strongly and frequently influenced by the*
*innervation zone [2, 3], in this case, neither the RMS or the mean frequency could be*
*monotonous. How to predict the muscle-tendon junction location?*

**Our response:** We appreciate the reviewer's meticulous and in-depth work. It is
accurate to say that ‘the innervation zone often and substantially influences the sEMG
signal pattern; hence, neither the RMS nor the mean frequency could be monotonous.’
Yet only when the sEMG is captured using a bipolar (single differential) arrangement
can this conclusion be made. As stated in ‘Part 2.1’ of Ref. 4 ‘... can be detected
using the monopolar or single differential (SD, bipolar) technique, ...’ the ‘bipolar’
and ‘single differential’ montage are the same, but distinct from the “monopolar.”
On closer examination of the 2 articles the reviewer provided, it was found that the
signals were identified in single differential mode in

188 A) the ‘*Method*’ section of Ref. 2 that ‘*The signals were detected in single*
*differential mode to minimize line interference, ...*’;
B) in the ‘*Equipment*’ section of Ref. 3 that ‘*... or 2.5 mm (silver pins 1 mm long,*
*1 mm diameter) (LISiN and OT Bioelettronica, Turin, Italy) in single*
*differential configuration.*’

In addition, as seen in Fig. 4 in Ref. 4, the amplitude of EMG recorded in monopolar
mode, decreases from the innervation zone to the tendon area, which is monotonous.
Based on this, we can identify the muscle-tendon junction location. We regret for
using the word ‘unipolar’ instead of ‘monopolar’ in the paper, which may have caused
some misunderstanding. Thus, on page 34, we make the following changes:

See Page 34: ‘**Bipolar recording was used for single-channel TPP electrode and**
**monopolar recording was used for MEAP to obtain sEMG signals.**’

*3. Why is value of 0.5 suitable as the muscle-tendon junction position, please explain*
*or add related reference.*

**Our response:** We appreciate the reviewer bringing this to our attention. We are sorry
that we cannot locate any relevant references because we are the first team to do
muscle-tendon junction localization using a sEMG electrode array, but this value of
0.5 was born with repeated verifications. When we used the MEAP to record on the
biceps, we discovered that various channels responded differently in mean frequency
(see Supplementary Fig. 16), and they consistently maintained a monotonous order
from tendon to muscle. We can plainly discern the junction site and its relative
position with our array using ultrasound images (the gold standard for tendon
monitoring) (see Fig. 7b). After comparing the results, we discovered that the junction
location was usually near to 0.5 after normalization (see Supplementary video 5).
In other words, we always saw the junction between two channels with normalized
values closer to 0.5 (see Fig. 7c). We further confirmed this rule by palpating the
biceps and Achilles tendon directly, and eventually established that 0.5 was the best
value to discern the junction point. After that, we tested this rule with palpation on
subjects 2 and 3. The outcomes were likewise comparable. We accept that this
method of locating may not be as exact as magnetic resonance imaging or ultrasound
imaging, but the convenience provided by our array should be underscored. Instead of

making this document excessively confusing, we decided to examine and summarize
a mature method or accuracy improvements in another study.

To clarify, we amended the phrase on page 22: ‘According to the results of ultrasound
image, we found the junction location was always close to the value of 0.5 after
normalization. We defined the channel with value of 0.5 is suitable as the muscle-
tendon junction position (Supplementary Text 1). This observation was also verified
by palpation on the biceps brachii and the Achilles tendon.’

*4. The inter-electrode distance could be largely changed during the isometric task*
*from flexion and extension due to the deformation of the sEMG array patch adhered*
*on muscle, how to define its effect?*

**Our response:** We appreciate the reviewer bringing this IED problem to our
attention. RMS and frequency can fluctuate as the IED between two electrodes
changes, especially in bipolar montage recording. Nevertheless, the recording setups
are monopolar for the majority of applications employing MEAP in this publication,
including identification of muscle loading, muscle exhaustion, and muscular activity
map. Signals from each electrode are unrelated to one another. The sole application
that may be affected is the location of the muscle-tendon junction, because the IEDs
are not constant during muscle activity, which may produce an inaccuracy in the
quantitative value of tendon displacements. However, we must underline that
changing the IEDs will not affect the detection of junctions using our normalizing
technique. Because the junction should always be located between two neighboring
channels, we may utilize the value of 0.5 to establish which two channels are
involved. In terms of the numerical value of tendon displacement, we have previously
studied this issue and modified computation to reduce the effect as much as feasible.
We added the demonstration in Supplementary Text 2.

See page 22: ‘We also improved calculation to make the influence of skin deformation
as low as possible (Supplementary Text 2).’

Supplementary Text 2:

We measured the IEDs between neighboring channels at the muscle-tendon junction
of the biceps distal tendon.

Flexion degree	IED ₂₁₋₁₇ (mm)	IED ₁₇₋₁₃ (mm)	IED ₁₃₋₉ (mm)	IED ₉₋₅ (mm)	IED ₅₋₁ (mm)
30°	15 mm	15 mm	15 mm	15 mm	15 mm
0°	15 mm	15 mm	16 mm	18 mm	18 mm
110°	15 mm	15 mm	14 mm	13 mm	12 mm

Since TPP electrodes are sticky and can adhere securely to the skin, the IED change is
the same as skin deformation between two nearby electrodes. We evaluated IEDs in
three distinct arm states: full extension (0°), full flexion (110°), and relax (30°). When
the array was bonded to the skin during arm relaxation, the IEDs were all 15 mm. We
discovered that when the muscle is moving, the deformations of skin on the muscle
part are not obvious.

For example, we assume the junction is right in the middle between channels 5 and 1
when the muscle is during full extension.

The calculated distance between junction and channel 21 is $D_{ce} = IED_{21-17}(30^\circ) + IED_{17-13}(30^\circ) + IED_{13-9}(30^\circ) + IED_{9-5}(30^\circ) + 1/2 IED_{5-1}(30^\circ)$;
the realistic distance between junction and channel 21 is $D_{re} = IED_{21-17}(0^\circ) + IED_{17-13}(0^\circ) + IED_{13-9}(0^\circ) + IED_{9-5}(0^\circ) + 1/2 IED_{5-1}(0^\circ)$ °

When the junction is right middle between channel 13 and 17 when full flexion,
the calculated distance between junction and channel 21 is $D_{cf} = IED_{21-17}(30^\circ) + 1/2 IED_{17-13}(30^\circ)$;
the realistic distance between junction and channel 21 is $D_{rf} = IED_{21-17}(110^\circ) + 1/2 IED_{17-13}(110^\circ)$.

Considering the absolute displacement D_a of channel 21 between flexion and
extension in the space,

then the calculated displacement is $D_{ce} - D_{cf} + D_a = 1/2 IED_{17-13}(30^\circ) + IED_{13-9}(30^\circ) + IED_{9-5}(30^\circ) + 1/2 IED_{5-1}(30^\circ) + D_a$;

the realistic displacement is $D_{re} - D_{rf} + D_a = 1/2 IED_{17-13}(0^\circ) + IED_{13-9}(0^\circ) + IED_{9-5}(0^\circ) + 1/2 IED_{5-1}(0^\circ) + D_a$;

The reason we choose distance between junction and channel 21 instead of channel 1
is that we found $D_a(21) \approx 0$ mm, while $D_a(1) \approx 40$ mm.

Calculating from the muscle end can therefore reduce the effect of significant skin
distortion.

Therefore, the computed displacement is 45 millimeters, but the real displacement is
50.5 millimeters. The inaccuracy is roughly 10%, which is deemed acceptable.
We did not see evident skin deformation between flexion and extension for Achilles
tendon identification, hence that we chose not to include this component in that
application.

Subject No.1's body fat percentage is roughly 20%, thus we don't observe many skin
deformations. Nevertheless, for persons with a body fat content of less than 15% or
even 10%, skin deformation can produce significant changes in IEDs and a higher risk
of non-conformal electrodes peeling off. In such instance, the stretchable electrode
array is more helpful and relevant, but it works best when combined with a strain
sensor on the patch to monitor and reduce the influence of skin deformation.
Similarly, we do not think these solutions should be discussed in this document to
ensure this manuscript is not too disorganized.

*5. For Figure 4g, we agree that decreasing median frequencies indicated fatigue of*
*the muscle. However, the authors claimed the decreasing value of slopes of median*
*frequencies indicated the fact that muscle became more fatigued. If so, the slopes of*
*median frequencies during each task (30 s) should not be a constant since the muscle*
*is getting fatigued as the time goes. Please have a check and related reference are*
*needed.*

**Our response:** We appreciate the reviewer bringing up the problem of fatigue. True,
the slope of median frequency does not remain constant during a task, especially if the
task duration is long. As time passes, the median frequency decline will be slower
because the source of tiredness is a reduction in the number of available motor units
to be innervated. We can see that the exponential regression may very well match the
median frequency reduction (See Fig. 5 of Ref. 5). The primary decline of median
frequency occurs in the 30s after the task starts. The issue is that utilizing the
exponential regression approach makes it difficult to compare each fatigue phase
quantitatively, but using linear fitting to get slope values allows us to do so. In fact,
using slope value is a common strategy to research muscle fatigue (Ref. 6).
In part 3.4 of Ref. 6, you can also see that the slopes of median frequency are lower
with increasing percentage of MVC, which corresponds to our statement in the
original manuscript that ‘the decreasing value of slopes recorded by TPP electrodes

also matched the fact that muscle became more fatigued after each task.’ Thanks to
the research mentioned by Reviewer 2, we can improve our new experiment design by
isolating the first 25 seconds of each task to undertake linear fitting to decrease error
as much as feasible. We appreciate the reviewer's thoughtful and considerate
comments, which will be extremely valuable for this work and future study.

See page 14: ‘To assess the ability of TPP electrodes to obtain information of
frequency in the signal, Ag/AgCl and TPP electrodes were set on the same position on
FCU, and the subject was asked to curl the wrist with a 5 kg dumbbell for three long
periods to activate FCU (Fig. 4e). The TPP electrodes showed a little better SNR than
the Ag/AgCl electrodes that they are 39.2, 37.5 and 40.5 dB for three contractions
recorded by TPP electrodes and 38.9, 37.5 and 38.6 dB by Ag/AgCl electrodes. The
spectrograms showed TPP electrodes can give clear frequency information just like
Ag/AgCl electrodes (Fig. 4f). To compare the performances of Ag/AgCl and TPP
electrodes on fatigue measurement, the subject was asked to curl the wrist 60 s for
three times for each type of electrodes. Three tasks were named as flexion 1, 2 and 3
to calculate median frequency during each task (Fig. 4g). Linear fittings were made
for first 25 s of each contraction, to quantify the outcome with less errors⁷³. The
slopes obtained by two types of electrodes both showed negative which indicated the
muscle was in fatigue. This test proved the TPP electrodes can measure the muscle
fatigue the same as Ag/AgCl electrodes.’

**Fig. 4 Comparison of recording performances on skin between Ag/AgCl and TPP**
 **electrodes.**

**a** Up, standard, compressing and stretching TPP electrodes on the skin. Scale bar: 1 cm;
 bottom, photographs of Ag/AgCl and TPP electrodes when recording sEMG of frontalis
 and the TPP electrode in the skin folds. Scale bar of photo at the bottom: 1 cm; bottom
 inset: 0.5 cm.

**b** sEMG signals recorded by Ag/AgCl and TPP electrodes, respectively. The subject
 was asked to make each contraction for 5 seconds. In the case of recording by Ag/AgCl
 electrodes, after four times of contraction, noises were even higher than signals.

**c** Schematic illustrations and lateral photos of Ag/AgCl electrode and TPP electrode on
 skin folds.

**d** Noise level and SNR recorded by two electrodes during contractions.

**e** Photographs of electrode configuration on FCU and contraction task. Two pairs of
 electrodes were attached the same position on the forearm.

**f** sEMG signals and spectrograms recorded by Ag/AgCl and TPP electrodes
 respectively.

**g** sEMG signals and fitting results of median frequency during flexion 1, 2 and 3
 recorded by Ag/AgCl and TPP electrodes. Decreasing median frequencies indicated

fatigue of the muscle.

*6. As shown in Figure 3d, TPP electrodes can work superbly with SNR level above 20*
*dB for almost 5 days, and then get worse. Please give the reason that decreases the*
*SNR of PPT electrodes for long-term measurement up to 5 days.*

**Our response:** The drop in SNR (caused by an increase in baseline noise level) of
commercial Ag/AgCl electrodes is caused by desiccation of conductive gels between
electrodes and skin. Many factors may contribute to a rise in baseline noise level in
TPP electrodes. We hypothesize that perspiration in normal life might gradually
destroy TPP electrodes because sweat fat and salts cannot escape and only slowly
accumulate on the TPP film. These interface changes can change the effective contact
area and conductivity of the film, increasing the noise intensity. Moreover, the
increase of the thickness of stratum corneum because of normal metabolism may
increase the impedance between electrodes and skin during such an extended period
of measurement. Nonetheless, TPP electrodes have been shown to have a
substantially longer effective use period than commercial gel electrodes.

See page 11: ‘As for the increase of baseline noise, we hypothesize that perspiration in
normal life might gradually the effective contact area and conductivity of the TPP
film because sweat fat and salts can only slowly accumulate on the TPP film,
increasing the noise intensity⁶⁸. Moreover, the formation increase of the thickness of
stratum corneum because of normal metabolism may increase the impedance between
electrodes and skin during such an extended period of measurement.’

*7. How about the adhesion, skin-electrode impedance and SNR of the sEMG electrode*
*array patch on the skin when after, e.g., 200 times, movements (compress or stretch)?*

**Our response:**

In response to this comment, we performed three tests to examine the change in
adhesion, impedance, and SNR after 200 times of compression or stretching. We
physically squeezed and stretched the skin with our fingertips to make the results
more convincing.

See page 8: 'Further, we found TPP electrodes showed excellent stability in adhesion,
skin-electrode impedance and SNR after 200 times of compress or stretch on the skin
(Supplementary Fig. 11, 12).'

**Supplementary Fig. 11 The change in adherence and impedance of TPP**
**electrodes on the skin.**

**a, b** images of the electrode applied to the skin and the motions made during the
adhesion test.

**c** The peeling force of TPP electrodes off the skin before and after motions.

**d, e** Images of the electrode applied on the skin and the process of movements during
impedance test.

**f** The impedance of TPP electrodes on the skin before, after motions and in the state
of compressing and stretching.

**Supplementary Fig. 12 SNR variation and baseline noise levels of TPP electrodes**
 **on the skin.**

**a, b** Images showing the electrode applied to the skin and the motions made to
 compare SNR.

**c** Demonstration of the entire process of recording using the TPP electrodes applied
 on skin before and after motions. The baseline noise level was reduced to 1.04 μV
 from 1.22 μV which was stable even after the motions. As a result, there was little
 change in the SNR of signals, showing the stability of TPP electrodes on the skin even
 after compression or stretching.

*8. Could be the sEMG electrode array patch be used repetitively?*

**Our response:** Using our TPP electrodes on the skin, we repeatedly performed
peeling-attaching studies and analyzed the noise of baseline to address this remark.
Even after 10 repetitions, the noise level barely changed. PPA layer on substrate may
also withstand several peelings and maintain enough adhesive force (Ref. 7). Hence,
we think the sEMG electrode array patch can be reapplied repeatedly.

See page 8: ‘TPP electrodes can also be used repetitively without changing the
baseline noise (Supplementary Fig.13).’

**Supplementary Fig. 13 The repetitive test of TPP films on the skin.**

*9. Please specific the thickness of each layer of the sEMG electrode array patch.*

**Our response:** We added the following sentence to The fabrication of MEAP.

See page 31: ‘The thickness of MEAPs was lower than 100 µm, typically with the
substrate of 65 µm, the encapsulation layer of 25 µm and the TPP electrode of 30 µm.’

*10. Bipolar recording was used for single-channel TPP electrode (Figure 4e) and
unipolar recording was used for MEAP, why and what’s the different?*

**Our response:** Bipolar recording can raise the SNR level to provide clearer
recordings by reducing shared noise between two electrodes. As the bipolar (single
differential) design is the one that is most commonly used in the EMG area, we
employed this mode for single-channel TPP electrodes to permit a direct comparison
with commercial Ag/AgCl electrodes. The SNR of monopolar recording is decreased
for applications employing MEAP, but it is still adequate for us to investigate
information about several muscles. In order to provide more convincing spatial
information, we thus create monopolar recordings to reduce the effects between each
electrode. This was covered in question 4 above.

We thank the reviewer again, for the thorough review and thoughtful advice. We
appreciated this precious opportunity to revise the manuscript according to the
reviewer's suggestions, which are extremely valuable to the improvement of our
manuscript.

*Reference*

[1]. Cao, J. et al. Stretchable and Self-Adhesive PEDOT:PSS Blend with High Sweat
Tolerance as Conformal Biopotential Dry Electrodes. *ACS Appl. Mater. Interfaces*
14, 39159–39171 (2022).

[2]. Farina D, Madeleine P, Graven-Nielsen T, et al. Standardising surface
electromyogram recordings for assessment of activity and fatigue in the human upper
trapezius muscle[J]. *European journal of applied physiology*, 2002, 86(6): 469-478.

[3]. Beretta Piccoli M, Rainoldi A, Heitz C, et al. Innervation zone locations in 43
superficial muscles: toward a standardization of electrode positioning[J]. *Muscle &*
*nerve*, 2014, 49(3): 413-421.

Reference for the response to the Reviewer #2

[1]. Cao, J. et al. Stretchable and Self-Adhesive PEDOT:PSS Blend with High Sweat
Tolerance as Conformal Biopotential Dry Electrodes. *ACS Appl. Mater. Interfaces*
14, 39159–39171 (2022).

[2]. Farina D, Madeleine P, Graven-Nielsen T, et al. Standardising surface
electromyogram recordings for assessment of activity and fatigue in the human upper
trapezius muscle[J]. *European journal of applied physiology*, 2002, 86(6): 469-478.

[3]. Beretta Piccoli M, Rainoldi A, Heitz C, et al. Innervation zone locations in 43
superficial muscles: toward a standardization of electrode positioning[J]. *Muscle &*
*nerve*, 2014, 49(3): 413-421.

[4] Merletti, R. & Muceli, S. Tutorial. Surface EMG detection in space and time: Best
practices. *J. Electromyogr. Kinesiol.* 49, 102363 (2019).

[5] Merletti, R. & Roy, S. Myoelectric and mechanical manifestations of muscle fatigue in
voluntary contractions. *J. Orthop. Sports Phys. Ther.* **24**, 342–353 (1996).

[6] Oliveira, A. de S. C. & Gonçalves, M. EMG amplitude and frequency parameters of
muscular activity: Effect of resistance training based on electromyographic fatigue threshold.
*J. Electromyogr. Kinesiol.* **19**, 295–303 (2009).

[7] Cheng, J. et al. Wet-Adhesive Elastomer for Liquid Metal-Based Conformal Epidermal
Electronics. *Adv. Funct. Mater.* 2200444 (2022) doi:10.1002/adfm.202200444.

**Reviewer #3 (Remarks to the Author):**

*The manuscript describes a surface electromyography electrode, which is novel, as*
 *the authors state, because it is characterised by the fact that it adheres adhesively to*
 *the skin surface, is stretchable and forms an array (see abstract). This claim by itself*
 *shows that the authors seem to be unfamiliar with the state-of-the-art in surface*
 *electromyography (sEMG). Adhesive sEMG electrodes that adhere independently to*
 *the surface of the skin have been available for several years. The manuscript does not*
 *comment on this, nor does it compare the supposedly so good new electrode with it.*

**Our response:** We appreciate the reviewer looking through our text and bringing up
 any issues. We consistently study the state-of-the-art in the sEMG area in order to
 evaluate the uniqueness of our work objectively. The TPP electrodes were compared
 with other good new electrodes, and it was mentioned on page 9: ‘In comparison with
 dry electrodes in reported literatures^{59–67}, TPP electrode performs better when the
 conformability and signal quality are evaluated (Fig. 2l, Supplementary Table 1, 2).’

**Supplementary Table 1 Comparisons between dry electrodes in other literatures and this**
 **work.**

	Materials for electrodes	Materials for substrates	Is it intrinsically stretchable?	Strain	The adhesiveness of electrode (N/cm)	The smallest area of the electrode (mm ²)	Electrode-skin impedance at 100 Hz (KΩ*cm ²)	Long-term test (Hour)	RM S of Noise (μV)	Signal-to-noise ratio (dB)	Reference
	Ag	Polyimide	No	80%	0	16.0	12.8	11	N/A	N/A	¹
	Ag-filled epoxy	Epoxy	No	N/A	0	100.0	80.0	24	~43.0	16.0	2
	Ag flakes/PDMS	PDMS	Yes	480%	0	100.0	34.0	10	~540.0	N/A	3
	Ag-polytetrafluoroethylene	Polyurethane	Yes	20%	0	600.0	N/A	N/A	~74.0	N/A	4
	Au nanoparticles	Polyimide	No	N/A	0	80.0	N/A	24	~60.0	~21.0	5
	PEDOT:PSS/Glycerol	Silk fiber	Yes	250%	N/A	314.0	~157.0	N/A	N/A	N/A	6
	PEDOT:PSS/Glycerol/Polysorbate	N/A	Yes	100%	0.013	100.0	200.0	12	N/A	35.2	7
	PEDOT:PSS/Polylactic acid	N/A	No	34%	~0.467	176.6	~35.3	N/A	~47.0	22.8	8
	PEDOT:PSS/Poly(poly(ethylene glycol) methyl ether acrylate)	N/A	Yes	75%	0.005	400.0	N/A	N/A	~60.6	4.5	9
	PEDOT:PSS/Polyvinyl alcohol/Borax	N/A	Yes	400%	N/A	254.3	101.7	N/A	N/A	29.5±1.3	10
	WPU/Deep eutectic solvent/Tannic acid	N/A	Yes	178%	0.125	1256.0	25	N/A	50.0	~14.0	11
	PEDOT/Waterborne polyurethane/D-sorbitol	N/A	Yes	43%	0.43	400	15	16	~25	~20	12
	PEDOT/Polyvinyl alcohol/Tannic Acid	N/A	Yes	54%	0.28	16	100	N/A	11.8	34.96	13
This work	PEDOT/Polyvinyl alcohol/Tannic Acid/Liquid metal	PDMS	Yes	188%	0.58	0.8	80	120	1.0	42.3±0.7	

One of these works was published in the year 2020, nine in 2021, and three in 2022.

We believe that this comparison accurately captures the current state-of-the-art in
 sEMG electrode technology. Just 6 out of 13 studies discuss sticky electrodes, and our
 TPP electrodes perform the best in terms of adhesiveness, which is important to note.
 TPP also fared the best in terms of long-term usage and signal quality. Our claim that
 our TPP electrodes are currently state-of-the-art is supported by this comparison.

Nevertheless, none of these electrodes could be used to create a stretchable array,
which drastically limited the number of applications that could be used in the sEMG
sector. In order to evaluate the effectiveness of our work objectively, we also
compared the performance of MEAP and other sEMG arrays. That part will be
discussed later for another concern about electrode arrays from the reviewer.
But we think the lack of detailed comparisons was the main reason for the concern
from the reviewer, so we added description about the comparison with other dry
electrodes.

See page 9: *'In the comparison, we also found only 6 out of 13 studies discussed*
*sticky electrodes, and our TPP electrodes perform the best in terms of adhesiveness,*
*which is an important contribution to its highest SNR among all dry electrodes.'*

*Instead, an unspecified Ag/AgCl electrode is used for comparison, which, as can be*
*seen from the figures, does not correspond to the standard for sEMG electrodes. It is*
*therefore doubtful to what the electrode introduced in the manuscript is compared to*
*and how meaningful this comparison is.*

**Our response:** We thank the reviewer for seeking clarification. We checked our
manuscript and found the details about the Ag/AgCl electrode was only mentioned in
the 'MATERIALS AND METHODS -- Impedance measurement'. The Ag/AgCl
electrode (Foam Monitoring Electrode 2228, 3M, USA) we used is a standard sEMG
electrode, and it has previously been reported in many articles by other researchers
(Ref. 14-18). So we believe all our comparisons are reliable and convincing. But we
also found the use of 2228 electrodes for sEMG just started in recent years, which
might not be recognized by the sEMG field. We took the advice from the reviewer
and bought Red Dot 2223 electrode from 3M for our experiments. 2223 electrodes
were used in literature far longer, so we think the results should be trustworthy (Ref.
19-25). This part will also be discussed below for another concern from the reviewer.

To make the experimental details clearer, we added details in 'MATERIALS AND
METHODS -- sEMG signal recording'.

See page 34: ‘Foam Monitoring 2228 electrodes were used for long-term test,
flexibility test on the forehead, and Red Dot 2223 (USA, 3M) electrodes were used
for fatigue tests.’

*Electrode arrays that adhere to the skin surface for long periods of time have also*
*been described since the 1990s and are now commercially available. There is no*
*reference to this in the manuscript either, nor is the introduced electrode compared to*
*them.*

**Our response:** We agree with the reviewer that this comparison between commercial
array (CA) and our MEAP is necessary. The material for the most popular
commercial sEMG array currently available is polyimide (PI), which has the Young's
modulus of 3 Gpa. Due to its characteristics, it was found that unless specific features,
such as serpentine design, were introduced, the array with PI substrate cannot make a
fully conformal contact with human skin (Young's modulus of 10 kPa). Also, we
recorded two movies to contrast the contact effectiveness of the PI sEMG array and
the MEAP on the skin (see Supplementary Video 2 and 3). Commercial array came
off either on the muscle or muscle-tendon junction during the muscle action, yet the
MEAP always maintained perfectly conformal contact. We also conducted sEMG
recording by CA and MEAP and added the results in the manuscript.

[revised manuscript text omitted]

We can see MEAP performs better than commercial arrays since a good contact is a
 crucial need for dependable and steady recording. We also compared flexible arrays
 described in the literature in Supplemental Table 2 of the original manuscript.

**Supplementary Table 2 Comparisons between sEMG arrays in other literatures and this**
 **work.**

Materials electrodes	for	Materials substrates	for	Is it intrinsically stretchable?	Strain	The adhesiveness of electrode (N/cm)	The number of channels	Success rate of channels	The smallest area of the electrode (mm^2)	Electrode-skin impedance at 100 Hz ($\text{K}\Omega\cdot\text{cm}^2$)	Long-term test (Hour)	RMS of Noise (μV)	Signal-to-noise ratio (dB)	Reference
Ag/AgCl ink		Polypropylene		No	N/A	0	16	N/A	5.1	5000.0	N/A	N/A	~24.0	26
Ag/AgCl ink		Polyethylene terephthalate		No	N/A	0	64	100%	14.5	N/A	2	N/A	~20.0	27
Ag flakes/PDMS		PDMS		Yes	30%	0	8	100%	26.4	33.0	N/A	N/A	29.5	28
Ag nanowires		PDMS		Yes	50%	0	18	94.4%	9.6	N/A	N/A	N/A	N/A	29
Ag nanowires		Thermoplastic polyurethane		Yes	600%	0	4	100%	201.0	1004.8	N/A	~34.0	26.6	30
Al		Polyethylene terephthalate		No	51%	0	16	100%	84.0	84.0	N/A	~130.0	N/A	31
Au		Polyimide		No	40%	0	20	N/A	N/A	N/A	N/A	~300.0	~20.0	32
Au		Polyimide		No	37%	0	64	N/A	0.8	117.0	N/A	~10.0	40.0 \pm 8.0	33
Au		Polyimide		No	N/A	0	64	N/A	3.1	N/A	N/A	N/A	26.0 \pm 6.0	33
Carbon/Silicone rubber		Textile		N/A	N/A	0	14	100%	400.0	320.0	N/A	~100.0	12.8 \pm 0.9	34
MXene		PDMS		No	N/A	0	40	100%	7.1	2.4	N/A	~34.0	N/A	35
MXene		Parylene-C		No	N/A	0	16	81.25%	2.6	256.0	N/A	~118.0	24.4 \pm 1.7	36
PEDOT:PSS/Choline lactate		Kapton		No	N/A	0	16	100%	2.6	45.0	N/A	~40.0	15.6	37
Stainless steel		Textile		No	N/A	0	150	90.6%	113.0	N/A	N/A	50.6 \pm 14.8	30.8 \pm 2.4	38
This work		PEDOT/Polyvinyl alcohol/Tannic acid/Liquid metal		Yes	188%	0.58	≥ 24	100%	0.8	80	120	1.0	42.3\pm0.7	

One of these works was released in 2020, nine were released in 2021, and three were
 released in 2022. We can see from this table that the only electrode having
 adhesiveness is ours. Also, the patch’s 188% strain is higher than that of the majority
 of prior works, further assuring conformal contact with the skin. For the lowest noise
 RMS for sEMG recording by MEAP in this comparison, both adhesiveness and
 stretchability are crucial. The straightforward construction and reliable operation of
 MEAP allow it to be employed for a variety of muscles and situations in addition to
 enhanced signal capture. Taking into account the foregoing discussion, we believe our
 work to be state-of-the-art in the sEMG array sector.

*This leaves the property of stretchability, which according to the authors should*
 *improve the quality of sEMG signals. The advantages and disadvantages of*

*stretchable electrode arrays have been debated among sEMG experts for many years.*
*The problem is, that the interelectrode distance changes when the array is stretched.*
*This affects the frequency spectrum of the sEMG signal in the case of a bipolar lead.*
*Investigations in the frequency domain, as suggested by the authors for fatigue*
*detection, are therefore not valid for non-isometric contractions, as it is not possible*
*to exclude beyond doubt that a measured change in the frequency domain is not due*
*to a change in the electrode distance. Stretchable electrode arrays are therefore*
*fundamentally unsuitable for such applications.*

**Our response:**

Yes, we concur with the reviewer's worry concerning the impact of IED changes on
sEMG signals. In bipolar mode of recording, it is true that changing the IED between
two electrodes can produce RMS and frequency changes. But the recording setups are
monopolar for most applications employing MEAP in this manuscript, including
identifying muscle loading, exhaustion, and muscular activity map. Signals from each
electrode are unrelated to one another. About the reviewer's concern concerning
fatigue assessment for non-isometric contractions, it is claimed that a reduction in
median frequency is also noticed via monopolar recording (Ref. 39). This was also
seen in our trials (see Fig. 8g). The variation in slope values under varied loads
revealed the muscle's various exhaustion stages. Extra figure below shows monopolar
recording data from the new fatigue trial described in Fig. 4e. Even though the signals
were heavily influenced by powerline noise and harmonics, the median frequencies
reduced during the contraction. All of these references and demonstrations lead us to
conclude that our stretchable electrode array is not only appropriate, but also offers an
unrivaled advantage for such applications due to its superior attachment performance
(see Supplementary Video 2 and 3).

**Extra figure for reviewer 3's comment.** Red dot 2223 3M electrodes and TPP
 electrodes were placed to identical positions on the FCU, and monopolar recording
 mode was chosen, and the median frequency was calculated. Both electrodes recorded
 decline in median frequency.

*This brings me to another problem concerning the manuscript. The manuscript is full*
 *of claims - often in the superlative - about signal quality and possible applications of*
 *the described electrode, which are not statistically proven. They seem to be the purely*
 *subjective perceptions of the authors. This becomes particularly clear in Fig. 2 k, in*
 *which a signal with a motion artefact, which occur from time to time but not*
 *regularly, was compared with the signal detected with the introduced electrode. The*
 *manuscript does not describe whether and if so how repeat measurements were*
 *carried out and how these were statistically evaluated to substantiate the statements*
 *made.*

**Our response:**

We thank the reviewer asking for clarification. The goal of Fig. 2k is to demonstrate
 that adhesive TPP electrodes are superior to our non-adhesive PEDOT-PVA electrode

for sEMG recording. We appreciate the reviewer's worry that motion artifacts occur
irregularly due to the wire-swinging effect. Nevertheless, the motion artefact in this
situation is mostly created by the relative movement of electrodes on the skin during
muscle movements. As can be seen in the Supplementary Fig. 8, the motion artifacts
are most severe during the biceps curl, whether recorded in monopolar or bipolar
mode. We also discovered that sticky TPP electrodes produce less motion artifacts
than non-adhesive PEDOT-PVA electrodes. We used statistical tools to examine 5
curls for each type of electrode and discovered that the RMS change differed between
them, proving our claim: the adhesive TPP electrodes are more suitable for sEMG
recording than our non-adhesive PEDOT-PVA electrode. About the reviewer's worry
about subjective conclusions, we accept that the lack of specifics in the experimental
portion contributed to this perception to some extent. Nonetheless, we need to point
out that all our findings are based on the outcomes of tests or comparative studies
with other publications. The SNR values in Fig. 2l were obtained from various
sources and compared. In Fig. 3d, three contractions were recorded every day to
provide statistical RMS and SNR values. In Fig. 8f, all RMS values collected
throughout the isometric exercise were used to create a box chart. In addition, we
tested three individuals in total to assess our MEAP performance on various persons
for injury prevention. We always strive to make things objective by using statistical
analysis of our findings.

We included information in 'sEMG signal recording' for better explanations to make
things clearer.

See page 34 'During the biceps brachii muscle recordings, a comparative test between
PEDOT-PVA and TPP electrodes was performed. The subject was instructed to keep
the curl speed between flexion and extension at 4.5 rad/s. Five contractions of each
electrode were tested for RMS alterations. For long-term test, three contractions were
recorded each day to provide statistical RMS and SNR values.'

*Fig. 4 a and e shows another problem that arises when characterising the quality of*
*the novel electrodes. The electrodes of the devices used for comparison are not*
*located in the same position as the novel electrodes. Rather, the comparison signals*
*are derived at less favourable positions, which has a negative influence on the signal*

*amplitude, the SNR and the frequency spectrum. An objective comparison between the*
*two devices is not possible under these conditions.*

**Our response:** We appreciate the reviewer's thoughtful comment on the experiment
design of this comparison test. We were also worried before that whether such
position difference caused difference in signal recording. The reason we chose to
record contractions by two types of electrodes simultaneously is we hope to get
'identical' signals and do analysis on the fatigue which should only be influenced by
the type of electrodes. We were more worried that each contraction itself would have
a difference in fatigue which introduced another variable into the analysis even if the
tasks were the same. So here is the issue of choice. We assumed remarkably close
positions of two electrodes would not make too much difference to the signal
recorded, so we chose that way originally. But we also agree with the reviewer's
opinion, so we conducted this experiment again to eradicate the position effect, trying
to make an objective comparison as the reviewer wished. We need to clarify that
through this new test, we cannot get the same conclusion that TPP electrodes are
better for fatigue measurements than Ag/AgCl electrodes, but we can say the TPP
electrodes can measure the muscle fatigue just as Ag/AgCl electrodes. It is worth
mentioning that we changed the Ag/AgCl electrodes from Foam Monitoring 2228 to
Red Dot 2223 electrodes in the new experiments. We updated Fig. 4 with the new
experiment.

See page 14: **'To assess the ability of TPP electrodes to obtain information of**
**frequency in the signal, Ag/AgCl and TPP electrodes were set on the same position on**
**FCU, and the subject was asked to curl the wrist with a 5 kg dumbbell for three long**
**periods to activate FCU (Fig. 4e). The TPP electrodes showed a little better SNR than**
**the Ag/AgCl electrodes that they are 39.2, 37.5 and 40.5 dB for three contractions**
**recorded by TPP electrodes and 38.9, 37.5 and 38.6 dB by Ag/AgCl electrodes. The**
**spectrograms showed TPP electrodes can give clear frequency information just like**
**Ag/AgCl electrodes (Fig. 4f). To compare the performances of Ag/AgCl and TPP**
**electrodes on fatigue measurement, the subject was asked to curl the wrist 60 s for**
**three times for each type of electrodes. Three tasks were named as flexion 1, 2 and 3**
**to calculate median frequency during each task (Fig. 4g). Linear fittings were made**
**for first 25 s of each contraction, to quantify the outcome with less errors⁷³. The**
**slopes obtained by two types of electrodes both showed negative which indicated the**

muscle was in fatigue. This test proved the TPP electrodes can measure the muscle
 fatigue the same as Ag/AgCl electrodes.'

76

4

Fig. 4 Comparison of recording performances on skin between Ag/AgCl and TPP electrodes.

**a** Up, standard, compressing and stretching TPP electrodes on the skin. Scale bar: 1 cm;
 bottom, photographs of Ag/AgCl and TPP electrodes when recording sEMG of frontalis
 and the TPP electrode in the skin folds. Scale bar of photo at the bottom: 1 cm; bottom
 inset: 0.5 cm.

**b** sEMG signals recorded by Ag/AgCl and TPP electrodes, respectively. The subject
 was asked to make each contraction for 5 seconds. In the case of recording by Ag/AgCl
 electrodes, after four times of contraction, noises were even higher than signals.

**c** Schematic illustrations and lateral photos of Ag/AgCl electrode and TPP electrode on
 skin folds.

**d** Noise level and SNR recorded by two electrodes during contractions.

**e** Photographs of electrode configuration on FCU and contraction task. Two pairs of
 electrodes were attached the same position on the forearm.

**f** sEMG signals and spectrograms recorded by Ag/AgCl and TPP electrodes
 respectively.

**g** sEMG signals and fitting results of median frequency during flexion 1, 2 and 3
 recorded by Ag/AgCl and TPP electrodes. Decreasing median frequencies indicated
 fatigue of the muscle.

*Finally, a comment on electrode arrays. The use of electrode arrays has been known*
*for a long time under the pseudonym High Density sEMG (HDsEMG) and is widely*
*used in different research questions. The method is called sEMG imaging and the*
*"heat maps" shown in Fig. 5 g and h are called "muscle activity maps" in the*
*literature. The fact that HDsEMG is suitable for localising anatomical structures*
*such as the neuromuscular junction or tendon insertion has been known since the*
*1990s and has been studied in a number of different investigations. In connection with*
*fatigue and pain, a change in the spatial distribution of the activity of the muscle has*
*already been demonstrated, as well as a change in the spectrum of the signal. This*
*fundamental work is not mentioned anywhere in the manuscript. Rather, the*
*impression is given that such investigations are only made possible by the new type of*
*electrode.*

**Our response:** We appreciate the reviewer's generous sharing about the knowledge
of HDsEMG. Yes, we agree the reviewer's point that HDsEMG is already widely
used in different research questions. The objective of showing our results in original
Fig. 5 is also to prove MEAP can complete the task carried before by HDsEMG, just
like we stated in the paragraph: 'Such tools will create remarkable benefits and
provide a new means for clinical diagnosis, medical treatment and sports sciences.'
The reason we said this is our stretchable array indeed can do those investigations
which our new type of electrode can complete better, but much harder by traditional
HDsEMG. We showed them in the new Supplementary Video 3 that traditional
HDsEMG is extremely easy to fall off when it is attached on the muscle-tendon
junction. So using traditional HDsEMG makes less stable recording for sEMG and let
alone the monitoring of tendon displacement under the skin.
As for the reviewer's concern on 'HDsEMG is suitable for localising anatomical
structures such as the neuromuscular junction or tendon insertion', we searched on
Pubmed (<https://pubmed.ncbi.nlm.nih.gov/>) with key words 'EMG', 'array' and
'tendon', there are only 33 results and none of them studied on the location of muscle-
tendon junction (Ref. 40-72), which is also different to two terms the reviewer
proposed. Based on that, we believe this application by MEAP is novel. We also used
it to help our injury prevention analysis which should prove that our stretchable
electrode array has potential for future applications in many areas such as clinical
diagnosis, medical treatment and sports sciences.

We changed ‘heatmaps’ to ‘muscle activity maps’ in manuscript.

See page 19: ‘Muscle activity maps based on RMS values were generated to visualize
the advantage of high-density systems.’

‘... which caused the active zone to move to the right in the muscle activity maps with
an increased activity.’

page 21: ‘g, h Muscle activity maps of sEMG recorded ...’

Page 34: ‘Muscle contraction task for muscle activity maps’, ‘Muscle activity maps
were generated to help visualize the change in activity during the task’.

For fundamental work about fatigue and pain, we added simple introduction in the
manuscript because this is not the focus of this manuscript.

See page 2: ‘There are also many works studied on neuromuscular junctions by high
density sEMG, to demonstrate the muscle fatigue and pain^{17–21}.’

17. Muceli, S. & Farina, D. Simultaneous and proportional estimation of hand
kinematics from EMG during mirrored movements at multiple degrees-of-
freedom. *IEEE Trans. Neural Syst. Rehabil. Eng.* **20**, 371–378 (2012).

18. Muceli, S., Falla, D. & Farina, D. Reorganization of muscle synergies during
multidirectional reaching in the horizontal plane with experimental muscle pain.
*J. Neurophysiol.* **111**, 1615–1630 (2014).

19. Merletti, R., Rainoldi, A. & Farina, D. Surface electromyography for
noninvasive characterization of muscle. *Exerc. Sport Sci. Rev.* **29**, 20–25 (2001).

20. Merletti, R., Farina, D. & Gazzoni, M. The linear electrode array: A useful tool
with many applications. *J. Electromyogr. Kinesiol.* **13**, 37–47 (2003).

21. Merletti, R. *et al.* Multichannel surface EMG for the non-invasive assessment of
the anal sphincter muscle. *Digestion* **69**, 112–122 (2004).

We thank reviewer’s incisive and thoughtful comments again for pointing out the issues
in the manuscript. We feel that our manuscript is more convincible, and the advantages
of MEAP are more unrivaled after we supplemented two experiments according to the
reviewer’s suggestions.

Reference for the response to the Reviewer #3

- 1. Harati, A. & Jahanshahi, A. A reliable stretchable dry electrode for monitoring of EEG signals.
*Sensors Actuators A Phys.* **326**, 112727 (2021).
- 2. Alban, M. V., Lee, H., Moon, H. & Yoo, S. Micromolding fabrication of biocompatible dry
micro-pyramid array electrodes for wearable biopotential monitoring. *Flex. Print. Electron.* **6**,
045008 (2021).
- 3. Jiang, Y. *et al.* Flexible and stretchable dry active electrodes with pdms and silver flakes for bio-
potentials sensing systems. *IEEE Sens. J.* **21**, 12255–12268 (2021).
- 4. Yoon, S. *et al.* Highly stretchable metal-polymer hybrid conductors for wearable and self-
cleaning sensors. *NPG Asia Mater.* **13**, 4 (2021).
- 5. Yun, I. *et al.* Stable Bioelectric Signal Acquisition Using an Enlarged Surface-Area Flexible Skin
Electrode. *ACS Appl. Electron. Mater.* **3**, 1842–1851 (2021).
- 6. Li, Q. *et al.* Highly Thermal-Wet Comfortable and Conformal Silk-Based Electrodes for On-
Skin Sensors with Sweat Tolerance. *ACS Nano* **15**, 9955–9966 (2021).
- 7. Tang, W. *et al.* Delamination-Resistant Imperceptible Bioelectrode for Robust
Electrophysiological Signals Monitoring. *ACS Mater. Lett.* **3**, 1385–1393 (2021).
- 8. Won, Y. *et al.* Biocompatible, Transparent, and High-Areal-Coverage Kirigami PEDOT:PSS
Electrodes for Electrooculography-Derived Human-Machine Interactions. *ACS Sensors* **6**, 967–
975 (2021).
- 9. Blau, R. *et al.* Intrinsically Stretchable Block Copolymer Based on PEDOT:PSS for Improved
Performance in Bioelectronic Applications. *ACS Appl. Mater. Interfaces* **14**, 4823–4835 (2022).
- 10. Zhou, X. *et al.* Self-healing, stretchable, and highly adhesive hydrogels for epidermal patch
electrodes. *Acta Biomater.* **139**, 296–306 (2022).
- 11. Wang, S. *et al.* Self-adhesive, stretchable, biocompatible, and conductive nonvolatile eutectogels
as wearable conformal strain and pressure sensors and biopotential electrodes for precise health
monitoring. *ACS Appl. Mater. Interfaces* **13**, 20735–20745 (2021).
- 12. Zhang, L. *et al.* Fully organic compliant dry electrodes self-adhesive to skin for long-term
motion-robust epidermal biopotential monitoring. *Nat. Commun.* **11**, 4683 (2020).
- 13. Cao, J. *et al.* Stretchable and Self-Adhesive PEDOT:PSS Blend with High Sweat Tolerance as
Conformal Biopotential Dry Electrodes. *ACS Appl. Mater. Interfaces* **14**, 39159–39171 (2022).
- 14. Roldan-Vasco, S., Restrepo-Agudelo, S., Valencia-Martinez, Y. & Orozco-Duque, A. Automatic
detection of oral and pharyngeal phases in swallowing using classification algorithms and
multichannel EMG. *J. Electromyogr. Kinesiol.* **43**, 193–200 (2018).

- 15. Zhuang, M. et al. Highly Robust and Wearable Facial Expression Recognition via Deep-
Learning-Assisted, Soft Epidermal Electronics. *Research* 2021, (2021).
- 16. Fortune, B. C., Pretty, C. G., Chatfield, L. T., McKenzie, L. R. & Hayes, M. P. Data captured
using low-cost active electromyography. *Data Br.* 29, 105239 (2020).
- 17. Becerra-Fajardo, L. et al. Floating EMG sensors and stimulators wirelessly powered and
operated
by volume conduction for networked neuroprosthetics. *J. Neuroeng. Rehabil.* 19, 57 (2022).
- 18. Ye-Lin, Y. et al. Directed Functional Coordination Analysis of Swallowing Muscles in Healthy
and Dysphagic Subjects by Surface Electromyography. *Sensors* 22, (2022).
- 19. Frenzel, D., Greim, C. A., Sommer, C., Bauerle, K. & Roewer, N. Is the bispectral index
appropriate for monitoring the sedation level of mechanically ventilated surgical ICU patients?
*Intensive Care Med.* 28, 178–183 (2002).
- 20. Campos, D. P. et al. Short-term fibre intake estimation in goats using surface electromyography
of the masseter muscle. *Biosyst. Eng.* 183, 209–220 (2019).
- 21. Phinyomark, A., Phukpattaranont, P. & Limsakul, C. Fractal analysis features for weak and
single-channel upper-limb EMG signals. *Expert Syst. Appl.* 39, 11156–11163 (2012).
- 22. Wernbom, M., Järrebring, R., Andreasson, M. A. & Augustsson, J. Acute Effects of Blood Flow
Restriction on Muscle Activity and Endurance During Fatiguing Dynamic Knee Extensions at
Low Load. *J. Strength Cond. Res.* 23, 2389–2395 (2009).
- 23. Allen, D. P. A frequency domain Hampel filter for blind rejection of sinusoidal interference
from
902 electromyograms. *J. Neurosci. Methods* 177, 303–310 (2009).
- 24. Lee, S. et al. Wireless Epidermal Electromyogram Sensing System. *Electronics* 9, 269 (2020).
- 25. Shim, J. K., Choi, H. S. & Shin, J. H. Effects of neuromuscular training on knee joint stability
after anterior cruciate ligament reconstruction. *J. Phys. Ther. Sci.* 27, 3613–3617 (2015).
- 26. Cantu, E. et al. Printed Multi-EMG Electrodes on the 3D Surface of an Orthosis for
Rehabilitation: A Feasibility Study. *IEEE Sens. J.* 21, 14407–14417 (2021).
- 27. Moin, A. et al. A wearable biosensing system with in-sensor adaptive machine learning for hand
gesture recognition. *Nat. Electron.* 4, 54–63 (2021).
- 28. Zhu, K., Guo, W., Yang, G., Li, Z. & Wu, H. High-Fidelity Recording of EMG Signals by
Multichannel On-Skin Electrode Arrays from Target Muscles for Effective Human-Machine
Interfaces. *ACS Appl. Electron. Mater.* 3, 1350–1358 (2021).
- 29. Lin, Y. et al. High-resolution and large-size stretchable electrodes based on patterned silver
nanowires composites. *Nano Res.* 15, 4590–4598 (2022).

30. Zhao, H. *et al.* Ultrastretchable and Washable Conductive Microtextiles by Coassembly of Silver

- Nanowires and Elastomeric Microfibers for Epidermal Human-Machine Interfaces. *ACS Mater.*
*Lett.* **3**, 912–920 (2021).
- 31. Gandla, S. *et al.* Ultrafast Prototyping of Large-Area Stretchable Electronic Systems by Laser
Ablation Technique for Controllable Robotic Arm Operations. *IEEE Trans. Ind. Electron.* **69**,
4245–4253 (2022).
- 32. Mei, Z., Zhao, N., Yang, B. & Liu, J. Flexible Concentric Ring Electrode Array for Low-Noise
and Non-Invasive Detection. in *Proceedings of the IEEE International Conference on Micro*
*Electro Mechanical Systems (MEMS)* vols 2021-January 266–269 (Institute of Electrical and
Electronics Engineers Inc., 2021).
- 33. Chandra, S. *et al.* Performance Evaluation of a Wearable Tattoo Electrode Suitable for High-
Resolution Surface Electromyogram Recording. *IEEE Trans. Biomed. Eng.* **68**, 1389–1398
(2021).
- 34. Alizadeh-Meghbrazi, M. *et al.* A Mass-Produced Washable Smart Garment with Embedded
Textile EMG Electrodes for Control of Myoelectric Prostheses: A Pilot Study. *Sensors* **22**, 666
(2022).
- 35. Driscoll, N. *et al.* MXene-infused bioelectronic interfaces for multiscale electrophysiology and
stimulation. *Sci. Transl. Med.* **13**, eabf8629 (2021).
- 36. Murphy, B. B. *et al.* A Gel-Free Ti₃C₂Tx-Based Electrode Array for High-Density, High-
Resolution Surface Electromyography. *Adv. Mater. Technol.* **5**, 2000325 (2020).
- 37. Velasco-Bosom, S. *et al.* Conducting Polymer-Ionic Liquid Electrode Arrays for High-Density
Surface Electromyography. *Adv. Healthc. Mater.* **10**, 2100374 (2021).
- 38. Ting, J. E. *et al.* Sensing and decoding the neural drive to paralyzed muscles during attempted
movements of a person with tetraplegia using a sleeve array. *J. Neurophysiol.* **127**, 2104–2118
(2021).
- 39. Kleine, B. U., Schumann, N. P., Stegeman, D. F. & Scholle, H. C. Surface EMG mapping of the
human trapezius muscle: The topography of monopolar and bipolar surface EMG amplitude and
spectrum parameters at varied forces and in fatigue. *Clin. Neurophysiol.* **111**, 686–693 (2000).
- 40. Carraro, U. Thirty years of translational research in Mobility Medicine: Collection of abstracts
of the 2020 Padua Muscle Days. *Eur. J. Transl. Myol.* **30**, 8826 (2020).
- 41. Huang, C., Chen, X., Cao, S. & Zhang, X. Muscle-tendon units localization and activation level
analysis based on high-density surface EMG array and NMF algorithm. *J. Neural Eng.* **13**, 66001
(2016).
- 42. Chen, X., Wang, S., Huang, C., Cao, S. & Zhang, X. ICA-based muscle-tendon units localization

- and activation analysis during dynamic motion tasks. *Med. Biol. Eng. Comput.* 56, 341–353
(2018).
- 43. Zhang, X. et al. EMG-Torque Relation in Chronic Stroke: A Novel EMG Complexity
Representation With a Linear Electrode Array. *IEEE J. Biomed. Heal. informatics* 21, 1562–
1572 (2017).
- 44. Thompson, C. K., Johnson, M. D., Negro, F., Farina, D. & Heckman, C. J. Motor Unit
Discharge
Patterns in Response to Focal Tendon Vibration of the Lower Limb in Cats and Humans. *Front.*
*Integr. Neurosci.* 16, 836757 (2022).
- 45. McAvoy, M. et al. Flexible Multielectrode Array for Skeletal Muscle Conditioning,
Acetylcholine Receptor Stabilization and Epimysial Recording After Critical Peripheral Nerve
Injury. *Theranostics* 9, 7099–7107 (2019).
- 46. Li, W. & Sakamoto, K. The influence of location of electrode on muscle fiber conduction
velocity and EMG power spectrum during voluntary isometric contraction measured with surface
array electrodes. *Appl. Human Sci.* 15, 25–32 (1996).
- 47. Merletti, R., Farina, D. & Gazzoni, M. The linear electrode array: a useful tool with many
applications. *J. Electromyogr. Kinesiol. Off. J. Int. Soc. Electrophysiol. Kinesiol.* 13, 37–47
(2003).
- 48. English, A. W., Wolf, S. L. & Segal, R. L. Compartmentalization of muscles and their motor
nuclei: the partitioning hypothesis. *Phys. Ther.* 73, 857–867 (1993).
- 49. Wakeling, J. M. The recruitment of different compartments within a muscle depends on the
mechanics of the movement. *Biol. Lett.* 5, 30–34 (2009).
- 50. Tokunaga, T. [Muscle fiber conduction velocity and frequency parameters of surface EMG
during fatigue of the human masseter muscle. 2. Frequency parameters]. *Nihon Hotetsu Shika*
*Gakkai Zasshi* 33, 804–817 (1989).
- 51. Farina, D. & Merletti, R. A novel approach for estimating muscle fiber conduction velocity by
spatial and temporal filtering of surface EMG signals. *IEEE Trans. Biomed. Eng.* 50, 1340–1351
(2003).

- 52. Thompson, C. K. et al. Robust and accurate decoding of motoneuron behaviour and prediction
of the resulting force output. *J. Physiol.* 596, 2643–2659 (2018).
- 53. Botter, A., Vieira, T. M., Geri, T. & Roatta, S. The peripheral origin of tap-induced muscle
contraction revealed by multi-electrode surface electromyography in human vastus medialis. *Sci.*
*Rep.* 10, 2256 (2020).
- 54. Merletti, R., Roy, S. H., Kupa, E., Roatta, S. & Granata, A. Modeling of surface myoelectric
signals--Part II: Model-based signal interpretation. *IEEE Trans. Biomed. Eng.* 46, 821–829
(1999).
- 55. Ye, X., Beck, T. W. & Wages, N. P. Prolonged passive static stretching-induced innervation
zone
shift in biceps brachii. *Appl. Physiol. Nutr. Metab. = Physiol. Appl. Nutr. Metab.* 40, 482–488
(2015).
- 56. De la Fuente, C. et al. Distal overactivation of gastrocnemius medialis in persistent
plantarflexion
weakness following Achilles tendon repair. *J. Biomech.* 148, 111459 (2023).
- 57. Minetto, M. A., Botter, A., Ravenni, R., Merletti, R. & De Grandis, D. Reliability of a novel
neurostimulation method to study involuntary muscle phenomena. *Muscle Nerve* 37, 90–100
(2008).
- 58. Bujalski, P., Martins, J. & Stirling, L. A Monte Carlo analysis of muscle force estimation
sensitivity to muscle-tendon properties using a Hill-based muscle model. *J. Biomech.* 79, 67–77
(2018).
- 59. Barandun, M., von Tscherner, V., Meuli-Simmen, C., Bowen, V. & Valderrabano, V.
Frequency
and conduction velocity analysis of the abductor pollicis brevis muscle during early fatigue. *J.*
*Electromyogr. Kinesiol. Off. J. Int. Soc. Electrophysiol. Kinesiol.* 19, 65–74 (2009).
- 60. Mito, K. & Sakamoto, K. On the evaluation of muscle fiber conduction velocity considering
waveform properties of an electromyogram in *M. biceps brachii* during voluntary isometric
contraction. *Electromyogr. Clin. Neurophysiol.* 42, 137–149 (2002).
- 61. Mananas, M. A., Rojas, M., Mandrile, F. & Chaler, J. Evaluation of muscle activity and fatigue

- Eng. Med. Biol. Soc. IEEE Eng. Med. Biol. Soc. Annu. Conf. 2005, 5824–5827 (2005).
- 62. Sakamoto, K. & Li, W. Effect of muscle length on distribution of muscle fiber conduction
velocity for M. biceps brachii. *Appl. Human Sci.* 16, 1–7 (1997).
- 63. Farina, D., Arendt-Nielsen, L., Merletti, R. & Graven-Nielsen, T. Assessment of single motor
unit conduction velocity during sustained contractions of the tibialis anterior muscle with
advanced spike triggered averaging. *J. Neurosci. Methods* 115, 1–12 (2002).
- 64. Rodriguez-Falces, J., Botter, A., Vieira, T. & Place, N. The M waves of the biceps brachii have
a stationary (shoulder-like) component in the first phase: implications and recommendations for
M-wave analysis. *Physiol. Meas.* 42, 15007 (2021).
- 65. Rojas, M., Mañanas, M. A., Muller, B. & Chaler, J. Activation of forearm muscles for wrist
extension in patients affected by lateral epicondylitis. *Annu. Int. Conf. IEEE Eng. Med. Biol.*
*Soc. IEEE Eng. Med. Biol. Soc. Annu. Int. Conf.* 2007, 4858–4861 (2007).
- 66. Mito, K. & Sakamoto, K. Distribution of muscle fiber conduction velocity of m. masseter
during
voluntary isometric contraction. *Electromyogr. Clin. Neurophysiol.* 40, 275–285 (2000).
- 67. Cho, S.-G. et al. Hand motion recognition based on forearm deformation measured with a
distance sensor array. *Annu. Int. Conf. IEEE Eng. Med. Biol. Soc. IEEE Eng. Med. Biol. Soc.*
*Annu. Int. Conf.* 2016, 4955–4958 (2016).
- 68. Masuda, T. & Sadoyama, T. The propagation of single motor unit action potentials detected by
a surface electrode array. *Electroencephalogr. Clin. Neurophysiol.* 63, 590–598 (1986).
- 69. Nishihara, K., Futami, T., Hosoda, K. & Gomi, T. Validation of estimated muscle fiber
conduction velocity with the normalized peak-averaging technique. *J. Electromyogr. Kinesiol.*
*Off. J. Int. Soc. Electrophysiol. Kinesiol.* 15, 93–101 (2005).
- 70. Li, W. & Sakamoto, K. Distribution of muscle fiber conduction velocity of M. biceps brachii
during voluntary isometric contraction with use of surface array electrodes. *Appl. Human Sci.*
15, 41–53 (1996).
- 71. Sakamoto, K. & Mito, K. Muscle fiber conduction velocity during isometric contraction and the
recovery period. *Electromyogr. Clin. Neurophysiol.* 40, 151–161 (2000).

- 72. Hník, P., Vejsada, R. & Macková, E. V. EMG activity in 'compensatory' muscle hypertrophy.
*Physiol. Bohemoslov.* 35, 285–288 (1986).

REVIEWER COMMENTS

Reviewer #1 (Remarks to the Author):

The authors have made well addressed my concerns in the revised manuscript. The quality of the revised version of this study is worthy of publishing in Nature Communcations.

Reviewer #2 (Remarks to the Author):

Here are some other comments which should be further addressed before its consideration for publication:

1. Could be the MEAP be used repetitively? (I mean the 4*6 array, not the PPT electrode)
2. How about the permeability of the MEAP since the authors claimed its long-term usage?
3. For the reader's better understanding, more details are expected on the MEAP connected to the data acquisition module via flexible printable circuit board connectors.
4. The authors compared the attachment performance between the commercial array and MEAP. In fact, the spatial density and distribution of the electrode sites is also important to its application. The commercial array has 64 channels with 8*8 array while the MEAP is 24 with 4*6 array, which leads to mismatch location of the electrode sites during comparison. It would be better to make a comparison directly with the same configuration.
5. In addition to the innovative aspects of the PPT electrode and the MEAP, please also specify the corresponding limitations at Conclusion Section.

Reviewer #3 (Remarks to the Author):

The authors have tried to address some of my concerns. However, they only succeeded to a very limited extent. The manuscript credibly demonstrates that the mechanical properties of the new electrode array are superior to other electrode arrays. The same applies to the purely electrical properties. However, the manuscript still has major shortcomings when the suitability of the new electrode array is examined with regard to the detection of sEMG signals and compared with state-of-the-art methods. As an example, I refer again to Fig. 4e, which still compares the sEMG signals derived with the new electrodes with signals from sEMG derivations that do not comply with the international recommendations for the derivation of sEMG signals. The international recommendations for the

detection of sEMG signals are defined and described in the SENIAM and CEDE projects. The project results are published as well as described on the internet. E.g. it seems, that the electrode distance is larger than 1/4 of the muscle fibre length of the FCU and that the sensor is placed halfway the (most) distal motor endplate zone and the distal tendon (SENIAM Recommendations). In addition, any information about the sensor used, such as size of the active area, interelectrode distance and exact position of the electrodes as well as the treatment of the skin, is missing. This information is essential to evaluate the quality of the derived sEMG signals and is standard in any sEMG publication, even when comparing two methods. Additionally, open cable clips were used to fix the cables to the electrodes. This worsens the SNR considerably. Shielded connectors are more common.

The manuscript is still full of claims - often in the superlative - about signal quality and possible applications of the described electrode, which are not statistically proven. I would like to draw special attention to the statistics here once again! I do not see any statistical calculation in the entire manuscript that shows that the claims made with respect to application are statistically significant, or that at least a tendency can be read. Instead, individual examples are shown, which is good, but does not justify the strong formulations and the emphasis on the new electrode over existing methods. That the major problem with the paper is the lack of statistics. As it stands, there is no evidence for the results in the different applications, and for me that is a no-go for a publication.

Furthermore, some of the claims about the state of the art of existing sEMG electrode/lead methods are simply wrong. Let me cite the abstract: "However, current sEMG electrodes do not offer adequate high-quality data for their widespread use in clinics and everyday life, since these are neither stretchable nor arrayed". This sentence is simply wrong! Electrode arrays have been available for many years and some of them are stretchable. Whether stretchability improves the clinical significance of the sEMG is unknown, while the clinical benefit of the array is proven. The manuscript is full of such examples and should be formulated in a less absolute way to reflect reality.

1 **Response to reviewers for the manuscript (NCOMMS-22-46103B-Z)**

2

*Reviewer #1 (Remarks to the Author):*

*The authors have made well addressed my concerns in the revised manuscript. The*
*quality of the revised version of this study is worthy of publishing in Nature*
*Communications.*

**Our response:** We appreciate the reviewer taking the time to carefully read our
revised

manuscript and provide such excellent feedback.

*Reviewer #2 (Remarks to the Author):*

*Here are some other comments which should be further addressed before its*
*consideration for publication:*

**Our response:** We really value the reviewer's time spent reading our revised
manuscript

thoroughly and providing other comments to further improve the manuscript.

1. *Could be the MEAP be used repetitively? (I mean the 4*6 array, not the PPT*

*electrode)*

**Our response:** We appreciate the reviewer seeking further clarification. In the previous
response letter, we proved TPP electrodes can be used repetitively by different tests. We
have also referenced our earlier publication¹, which demonstrates the reattachment of
the substrate part on the skin repetitively. By combining these findings, we aimed to
convey that the MEAP is suitable for repetitive use.

To provide a more convincing answer to the reviewer's question, we have designed a
new experiment specifically to demonstrate the repetitive use of the MEAP on the skin.
However, it should be noted that the signal qualities obtained from this experiment were
slightly lower compared to previous tests conducted using TPP electrodes. This is
primarily due to the much smaller contact area of the channels in the MEAP.
Nevertheless, the signal qualities obtained were still sufficiently high (approximately
20 dB) for use after repetitive attachment.

While we acknowledge the value of the experiment suggested by the reviewer, we have
decided that incorporating these results into the main body of the text would disrupt the
current outline, as the manuscript primarily focuses on electrodes and the MEAP
separately. Therefore, we have chosen to include these additional findings in the
Supplementary Information section.

Our changes on Page 11: **As for the MEAP, we checked the reattachment performance**
**of the patch (Supplementary Fig. 14). The results showed all channels of MEAP have**
**stable performances. This indicates that MEAP can be used repetitively.**

**Supplementary Fig. 14 The reattachment test of 24-channel MEAP on the skin.**

**a** Images showing the process of detachment and reattachment of MEAP on biceps
brachii.

**b, c** The baseline noise of each channel and the SNR of each channel plotted for each
reattachment. The baseline noise consistently maintained an amplitude of
approximately 50 μV across all channels, even after 28 reattachments. Similarly, the
SNR remained stable at 20 dB across all channels after 28 reattachments.

**d, e** Whisker plots of statistical verification of **b** and **c** for the 28 reattachments.
Statistical analysis was conducted to assess the baseline noise and SNR of each channel
throughout the reattachment test. 28 reattachments, per channel, were included in the
analysis. The box plots depict the mean (center square), median (center line), 25th to

75th percentiles (box), and the lower and upper whiskers representing the smallest and
largest values that are ≤ 1.5 times the interquartile range, respectively. Outliers are also
shown.

*2. How about the permeability of the MEAP since the authors claimed its long-term*
*usage?*

**Our response:** We appreciate the reviewer for highlighting the concern regarding
permeability. The insensible sweat rate of individuals typically ranges from 12 to 42

$\text{g}\cdot\text{m}^{-2}\cdot\text{h}^{-1}$ (Ref. 2), indicating that materials with similar permeability can meet the
requirements for daily use or long-term wearing. The substrate material used for the
MEAP is PDMS, which inherently possesses permeability. Our results demonstrate
that
the MEAP has a permeability of approximately $20 \text{ g}\cdot\text{m}^{-2}\cdot\text{h}^{-1}$, which is suitable for the
normal evaporation of sweat.

Moreover, if necessary, punctures can be made on the substrate to further enhance the
permeability of the MEAP. This adjustment can be customized based on individual
experiments and subject requirements. However, it is important to note that in
comparison to PDMS, polyimide exhibited significantly lower permeability, implying
limitations for long-term use.

As per the structure of our manuscript, similar to the first additional test, we have
included these results in the Supplementary Information section. This decision allows
92 us to maintain the coherence and flow of the main body text.

Our modifications on Page 11: **We also examined the permeability performance of**
**MEAP for daily long-term use (Supplementary Fig. 15). The results demonstrate that**
**the permeability of the MEAP is well-suited for extended periods of usage, as it does**
**not hinder the normal evaporation of sweat from the skin. Furthermore, we discovered**
**that the permeability of the MEAP can be adjusted by modifying the physical structure**
**of the substrate. This ability to tune the permeability enables us to create a comfortable**
**wearing experience for daily use, as the permeability can be increased to a level that**
**promotes adequate airflow.**

**Supplementary Fig. 15 The permeability comparison test between MEAP, MEAP**
 **(punctures) and polyimide.**

**a, b** Images showing the MEAP and MEAP (punctures) with thicknesses of 80 and 87
 106 μm , respectively. The MEAP (punctures) features 24 punctures (1 mm in diameter),
 corresponding to the number of TPP electrodes on the patch.

**c** The experimental setup for the permeability test. Three beakers, each filled with 100

109 ml of deionized water, were covered by MEAP, MEAP (punctures), and polyimide,
respectively. Each beaker was secured with a rubber band to ensure that water only
passed through the cover. The three beakers were placed in a programmable
temperature and humidity tester (QHP-360BE, LICHEN, China) set to a temperature
of 33 °C and a humidity of 30%, simulating the conditions on human skin.

**d** The water loss rates in the three beakers. The MEAP (punctures) exhibited higher
water loss compared to the MEAP, indicating that the permeability can be adjusted by
modifying the physical structure of the substrate. Measurements were recorded for each
beaker every hour, with $n = 3$ samples for each recording.

**e** The water loss rate of each cover. Considering that the insensible sweat rate of
individuals ranges from 12 to 42 $\text{g}\cdot\text{m}^{-2}\cdot\text{h}^{-1}$, the permeability of the MEAP is sufficient
to provide a comfortable wearing experience for daily use.

*3. For the reader's better understanding, more details are expected on the MEAP*
*connected to the data acquisition module via flexible printable circuit board connectors.*

**Our response:** We appreciate the reviewer's request for more details regarding the
connection. In our study, we employed hot-pressing to combine the Flexible Printed
Circuit (FPC) with the MEAP. By using a customized back-end connector, each channel
of the MEAP could be independently connected to the G.tec recording system. For
better understanding, we have included a photograph in the Supplementary Information,
illustrating the entire setup.

**Supplementary Fig. 20 The whole setup for connection between MEAP and EMG**
**recording system.**

See page 16:

The MEAP was connected to the sEMG recording system via flexible printed circuit
(Supplementary Fig. 20).

We also added more information in the MATERIALS AND METHODS. See page 32:

Note that electrode sites and connection pads were protected by silicone films during
the encapsulation to allow the electrodes and connection pads to be exposed.

The front-end connectors were made by polyimide flexible printed circuit (FPC). Front-
end connectors were designed and fabricated (EasyEDA, China) for connecting specific
MEAPs. The FPC and MEAP were hot-pressed together with force of 50 N and
temperature of 140 °C for 30 s by a hot-pressing machine (G311, Freamc, China). With
a customized back-end connector, every channel of MEAP can be independently
connected to EMG recording system.

*4. The authors compared the attachment performance between the commercial array
and MEAP. In fact, the spatial density and distribution of the electrode sites is also
important to its application. The commercial array has 64 channels with 8*8 array
while the MEAP is 24 with 4*6 array, which leads to mismatch location of the electrode
sites during comparison. It would be better to make a comparison directly with the same
configuration.*

**Our response:** We appreciate the reviewer's insightful comment, and we highly value
the advice provided. We agree with the reviewer's suggestion regarding the
configuration of the electrode array and its potential impact on the recording results. To
ensure a fairer comparison, we repeated experiment using the updated 64-channel
MEAP. This updated MEAP has the exact same configuration as the commercial
electrode array, including the surface area of the substrate (Supplementary Fig. 19). We
applied the same analysis method to process the data obtained from the experiment, and
the results consistently demonstrated that the 64-channel MEAP exhibited superior and
more stable performance in terms of baseline noise and SNR when compared to the
commercial electrode array, both on muscle and muscle-tendon junction recordings.
Although the smaller contact area of the electrode on the 64-channel MEAP led to a
slightly lower SNR compared to the previous 24-channel MEAP, the differences in
performance between the MEAP and the commercial electrode array remained

statistically significant. Given these findings, most of our previous conclusions remain
 valid, and we have made minimal changes to the text. The main changes have been
 implemented in Figure 5 and two Supplementary Videos.

**Fig. 5 The comparison of attachment performances between commercial array**
 **and MEAP.**

**a, e** Photographs of CA and MEAP attachment, illustrating the difference when

recording from muscle and muscle-tendon junction of biceps brachii.
**b, f** sEMG signals recorded using CA and MEAP on muscle and muscle-tendon junction
of biceps brachii. Four typical channels were picked for each recording.
**c, g** Spatial SNR performance map for each channel of CA and MEAP for the first and
last muscle contraction. SNR_f: SNR of the first contraction; SNR_l: SNR of the last
contraction.
**d, h** Statistical analysis of performances between CA and MEAP, including baseline
noise level of CA before and after one or three muscle contractions, as well as after
reattachment; baseline noise level of MEAP before and after ten muscle contractions;
baseline noise change rates before and after muscle contractions; SNR performance of
the last muscle contraction recorded by each of the CA and MEAP channels.
Significance was determined by one sample t test (*P< 0.05; **P< 0.01; ***P< 0.001;

****P < 0.0001).

**Supplementary Fig. 19 The configuration comparison between MEAP and CA.**

**Both arrays have electrode diameter of 4 mm and IED of 8 mm.**

See page 16: ...with human skin (Young's modulus of 10 kPa). **To fairly compare the**
**contact performance on the skin, we fabricated a 64-channel MEAP with the same**
**configuration as the CA (Supplementary Fig. 19).** We recorded a movie to ... resulting
in a significantly lower SNR. **While the MEAP exhibited a much more stable noise**
**level even after ten contractions, maintaining a high SNR.** We also recorded sEMG
signals... the first contraction to the last one (Fig. 5g). **MEAP on the other hand,**
**produced stable recordings with all channels having SNR greater than 15 dB.** Statistical
analysis of sEMG...

*5. In addition to the innovative aspects of the PPT electrode and the MEAP, please also*
*specify the corresponding limitations at Conclusion Section.*

**Our response:** We appreciate the reviewer's suggestion for a more comprehensive
discussion of our work. We acknowledge that there are certain limitations to the current
MEAP design, and we are actively working towards addressing them to expand the
potential applications of MEAP. One specific area of focus is the accurate recording
and recognition of single motor unit signals from surface electromyography (sEMG)

signals. If MEAP can achieve this capability, it has the potential to be utilized in clinical
diagnosis, replacing the use of needle electrodes. This would offer patients a much more
comfortable experience while maintaining the accuracy and reliability of the diagnostic
process. Such a development could revolutionize EMG clinical diagnosis by replacing
invasive tools with non-invasive alternatives. Furthermore, we recognize that the cable
connection is currently a limiting factor in the daily application of MEAP. We are
actively addressing this issue to improve the overall user experience and make MEAP
more suitable for everyday use. We have incorporated these discussions into the
conclusion section of our manuscript to provide a more comprehensive overview of the
potential implications and future directions of our work.

See page 30: e.g., prosthetics or virtual reality. **We also aim to replace invasive needle**
**electrodes in clinical applications with MEAP to provide patients with improved**
**comfort. However, current MEAPs have difficulty recognizing single motor unit signals**
**from sEMG recording because the lack of intelligent back-end algorithms causes low**
**single motor unit selection efficiency and accuracy. In this case, MEAP can only**
**provide limited help to clinical diagnosis. Another drawback of current MEAPs is they**
**are using cable connection, which limits the application scenarios and simplicity. To**
**address these issues, we are endeavoring to combine MEAP with intelligent algorithms**
**and wireless modules to make whole devices more useful in clinic scenario and more**
**portable in daily life. We believe in the future, MEAP has enormous potential to be**
**commercialized because of its low cost and simple fabrication, thus providing a new**
**platform for disease diagnosis, daily rehabilitation management and scientific exercise.**

*Reviewer #3 (Remarks to the Author):*

*The authors have tried to address some of my concerns. However, they only succeeded*
*to a very limited extent. The manuscript credibly demonstrates that the mechanical*
*properties of the new electrode array are superior to other electrode arrays. The same*
*applies to the purely electrical properties.*

**Our response:** We sincerely appreciate the reviewer's acknowledgement of our effort
and the recognition of the superior mechanical and electrical properties of our electrode
array. We are grateful for the time and attention the reviewer dedicated to carefully
reviewing our manuscript once again.

*However, the manuscript still has major shortcomings when the suitability of the new*
*electrode array is examined with regard to the detection of sEMG signals and compared*
*with state-of-the-art methods. As an example, I refer again to Fig. 4e, which still*
*compares the sEMG signals derived with the new electrodes with signals from sEMG*
*derivations that do not comply with the international recommendations for the*
*derivation of sEMG signals. The international recommendations for the detection of*
*sEMG signals are defined and described in the SENIAM and CEDE projects. The*
*project results are published as well as described on the internet. E.g. it seems, that the*
*electrode distance is larger than 1/4 of the muscle fibre length of the FCU and that the*
*sensor is placed halfway the (most) distal motor endplate zone and the distal tendon*
*(SENIAM Recommendations).*

**Our response:** We thank the reviewer for the criticism about the sEMG derivation in
our manuscript. We appreciate your feedback as it helps us improve the article. Upon
thorough examination of our sEMG derivation method and the articles the reviewer
suggested, we found that the majority of our protocols align with international
recommendations. Allow us to address each point in detail:

We have carefully compared two standards proposed by the reviewer with our
electrodes based on the SENIAM guidelines. We apologize for the omission of the
Inter-electrode distance (IED) information in the 'sEMG signal recording' section,
where the IED between two Red Dot 2223 or TPP electrodes is consistently set at 20
272 mm. This oversight may have led the reviewer to perceive that '*the electrode distance*
*is larger than 1/4 of the muscle fiber length of the FCU*'. However, after reviewing

several articles investigating the muscle length of the Flexor Carpi Ulnaris (FCU), we
found that our IED of 20 mm is actually less than 1/4 of the typical muscle fiber length^{3,4}.
For instance, Table 1 in Ref 3 demonstrates that the FCU muscle length is '**236.5 ± 5.4**
**mm**'.³ Additionally, Ref 4 provides the details that '*The flexor carpi ulnaris muscle*
*presents a total length of 26.5 cm, with a muscular belly of 24.5 cm of length by 3.5*
*cm of width and 0.5 cm of thickness;*'⁴ These findings confirm that the usual length of
the FCU exceeds 200 mm, indicating that our IED of 20 mm falls within the acceptable
range of being less than 1/4 of the muscle fiber length of the FCU.

Regarding electrode placements, SENIAM recommends the sensor be placed halfway
between the (most) distal motor endplate zone and the distal tendon. However, the
endplate zone is hard to identify for each muscle, so researchers usually attach the
electrodes on the muscle belly or the bulkiest part of the muscle⁵, which is also reported
by SENIAM. We opted to follow this guidance because we did have difficulty in
locating the endplate zone precisely. We supplement this information in our method
section.

See page 34: '**For fatigue comparison tests, electrodes were attached on the most**
**prominent bulge of the muscle belly of FCU, and different types of electrodes were**
**attached on the exactly same position.'**

To provide further clarification that we followed the SENIAM recommendations for all
our sEMG derivations, we present the evidence below:

Electrode Shape and Size: As we compared our electrodes with the commercial Red
Dot 2223 electrodes, the shape and size parameters were predetermined. However, we
have previously discussed the reliability of these electrodes in our response letter,
ensuring that this aspect aligns with the standard.

Inter-Electrode Distance (IED): '*SENIAM recommends to apply the bipolar sEMG*
*electrodes around the recommended sensor location with an inter electrode distance*

*of 20 mm.* We have followed the recommendation and adjusted the IED to 20 mm
accordingly.

Orientation of Electrodes: *‘SENIAM recommends that the bipolar SEMG electrodes*
*should be placed around the recommended sensor location with the orientation*
*parallel to the muscle fibres.* We have carefully followed the recommendation and
aligned the orientation of our electrodes parallel to the FCU fiber.

Fixation on the Skin: *‘SENIAM recommends to use elastic band or (double sided)*
*tape / rings for the fixation of the electrodes(construction) and cables to the skin in*
*such a way that the electrodes are properly fixed to the skin, movement is not*
*hindered and cables are not pulling the electrodes(construction).* Our Ag/AgCl and
TPP electrodes were properly fixed to the skin because they are both sticky.

Location of the Reference Electrode: *‘Depending on the application SENIAM*
*recommends to use the wrist, the proc. spin. of C7 or the ankle as the standard*
*location of the reference electrode.* In our study, we opted to attach the reference
electrode to the elbow to maintain a stable reference potential.

With regard to CEDE, we inspected all related published matrices⁶⁻¹⁰ but could not find
any details about sEMG derivations. Instead, we used the article¹¹ by Hermens et al
([https://doi.org/10.1016/S1050-6411\(00\)00027-4](https://doi.org/10.1016/S1050-6411(00)00027-4), with >6000 citations on Google scholar)
which is cited by the CEDE matrices and provides details about sEMG derivations, to
compare with our own sEMG recording protocols. Allow us to compare each point in
detail:

Electrode material: *‘For bipolar or monopolar electrodes, it is obvious that Ag/AgCl*
*was the preferred electrode material.* *‘It is recommended to use pre-gelled Ag/AgCl*
*electrodes.* In our manuscript, the Red Dot 2223 electrodes (pre-gelled Ag/AgCl
electrodes) were used.

Electrode shape and size: *'Thus, in the literature both rectangular (bars) and circular*
*electrodes are being used for SEMG recordings of which circular electrode are by*
*far the most used.'* In our manuscript, the Red Dot 2223 electrodes are pre-gelled and
have a surface area of 2.4 cm². To make a fair comparison, our TPP electrodes fabricated
with the same surface area were used.

Inter-electrode distance: *'Authors seem to have a preference for IED values which*
*are a multiple of 10 mm. The largely preferred distance was 20 mm.'* We have
followed the recommendation and adjusted the IED to 20 mm accordingly.

Skin preparation: *'In the remaining 76 (53%) publications, standard skin preparation*
*techniques were mentioned such as shaving, rubbing/abrasion and cleaning of the*
*skin, or a combination of these techniques.'* In fact, one advantage of our TPP
electrodes is that they require no skin preparation or gel, as their soft adhesive properties
allow excellent contact with the skin, even in the presence of hair. Although feasible,
we did not use skin preparation or gel for both the commercial Ag/AgCl and TPP
electrodes to ensure a fair comparison. We specifically selected a less-hairy position on
the arm, which is the FCU, to obtain higher signal quality.

Sensor location and orientation on the muscle: *'Globally, three placement strategies*
*can be discerned: 1. on the center or on the most prominent bulge of the muscle belly*
*(10 out of 21); 2. somewhere between the innervation zone and the distal tendon (6*
*out of 21); 3. on the motor point (1 out of 21).'* Compared to SENIAM, the most
prominent bulge of the muscle belly was also mentioned for electrode placement in this
article. This is the exact location we selected for our comparison tests between Ag/AgCl
and TPP electrodes on FCU, because we are not able to figure out the precise location
mentioned in the second strategy. And the first strategy (on muscle belly) was most
used by other researchers, making us believe our sEMG derivations accords with
international standards.

In our later sEMG derivations using the MEAP, we adopt a high-density recording
approach instead of the conventional bipolar recording method. This approach allows
367 us to capture sEMG signals using our novel array patch, which is not currently available
on the market. As a result, it may not always be applicable or appropriate to utilize
established standards designed for conventional tools when designing protocols for our
unique tool. But we still follow the fundamental mechanism of sEMG and some high-
density recording rules¹² to set our protocols, such as the configuration of array, the
attachment position on the muscle and particular tasks for sEMG recording. We believe
the reviewer will understand and appreciate our effort.

*In addition, any information about the sensor used, such as size of the active area,*
*interelectrode distance and exact position of the electrodes as well as the treatment of*
*the skin, is missing. This information is essential to evaluate the quality of the derived*
*sEMG signals and is standard in any sEMG publication, even when comparing two*
*methods*

**Our response:** We thank the reviewer asking more details about all EMG recording
process. Upon reevaluation of our manuscript, we have identified that some of the
information, although included, was not presented clearly. We agree with the reviewer
that this information is critical for sEMG publication, and we have summarized these
below.

Size of the active area: we mentioned them in the experiment section ‘**Impedance**
**measurement**’ on page 34, but we also found the description in ‘**sEMG signal**
**recording**’ is missing, so we added the information to this part.

See page 34: ‘2228 and 2223 electrodes have the surface contact area of 2 and 2.4 cm².

The TPP electrodes in each comparison test have the same surface contact area with
2228 or 2223 electrodes correspondingly.’

Interelectrode distance:

We added the detailed information to the 'sEMG signal recording' part.

See page 34: 'All interelectrode distance for bipolar recording is 20 mm.'

Position of the electrodes:

We added the detailed information to the 'sEMG signal recording' part.

See page 34: 'Foam Monitoring 2228 electrodes were used for long-term test, flexibility
test on the forehead, and Red Dot 2223 (USA, 3M) electrodes were used for fatigue
comparison tests. For fatigue comparison tests, electrodes were attached on the most
prominent bulge of the muscle belly of FCU, and different types of electrodes were
attached exactly on the same position.'

Treatment of the skin:

As mentioned in the previous part, no needed treatment of the skin is an advantage of
our electrodes. Thus, we did not use any treatment of the skin in all sEMG tests through
our manuscript.

See page 35: 'For all sEMG recording, no skin treatment was used, including shaving,
rubbing or cleaning of the skin.'

We thank again the reviewer helping us supplement more detailed information to the
manuscript for readers to understand clearer.

*Additionally, open cable clips were used to fix the cables to the electrodes. This worsens*
*the SNR considerably. Shielded connectors are more common.*

**Our response:** We acknowledge the reviewer's viewpoint that shielded connectors are
more commonly used and preferable for eliminating factors that may degrade the SNR.
However, we encountered difficulties in finding a shielded connector on the market that
would be compatible with our single-channel TPP electrodes. Developing a custom
shielded connector would have been costly and time-consuming. As a compromise, we

opted to use crocodile clips as connectors and employed tape fixation to mitigate SNR
reduction as much as possible. It is worth noting that several articles published in
reputable journals have also used open cables, even at the expense of sacrificing SNR
values¹³⁻¹⁵. In this comparison test, our objective is to assess the performance of the
electrodes rather than solely pursuing the highest SNR. Thus, ensuring a fair
comparison between the two electrode types is our primary focus. As a result, we have
used open cables for both the Ag/AgCl electrodes, based on the same connector used
for the TPP electrodes. This decision was made to ensure consistency and fairness in
our comparison. We hope that the reviewer understands our rationale behind using open
cables for both electrode types.

*The manuscript is still full of claims - often in the superlative - about signal quality and*
*possible applications of the described electrode, which are not statistically proven. I*
*would like to draw special attention to the statistics here once again! I do not see any*
*statistical calculation in the entire manuscript that shows that the claims made with*
*respect to application are statistically significant, or that at least a tendency can be*
*read. Instead, individual examples are shown, which is good, but does not justify the*
*strong formulations and the emphasis on the new electrode over existing methods. That*
*the major problem with the paper is the lack of statistics.*

**Our response:** We appreciate the reviewer for bringing up the lack of statistical analysis
in our manuscript. We carefully reviewed our text and found that there was only one
instance of a superlative expression found on page 9: "***In the comparison, we also***
***found only 6 out of 13 studies discussed sticky electrodes, and our TPP electrodes***
***perform the best in terms of adhesiveness, which is an important contribution to its***
***highest SNR among all dry electrodes.***" We provided Figure 2I as supporting evidence,
and all references can be found to support this statement. Thus, we believe that the
superlative expression used in this context is appropriate. With regard to the signal
quality comparison, we agree with the reviewer's suggestion that the inclusion of
statistical analysis would strengthen our results. We have now incorporated statistical

analysis into Figure 5d and h. This addition provides more robust evidence to support
 the conclusion that MEAP demonstrates superior performance compared to commercial
 arrays in terms of signal quality and stability.

See page 18:

**Fig. 5 The comparison of attachment performances between commercial array**
 **and MEAP.**

**a, e** Photographs of CA and MEAP attachment, illustrating the difference when
 recording from muscle and muscle-tendon junction of biceps brachii.

**b, f** sEMG signals recorded using CA and MEAP on muscle and muscle-tendon junction
 of biceps brachii. Four typical channels were picked for each recording.

**c, g** Spatial SNR performance map for each channel of CA and MEAP for the first and
 last muscle contraction. SNR_f: SNR of the first contraction; SNR_l: SNR of the last
 contraction.

**d, h** Statistical analysis of performances between CA and MEAP, including baseline
 noise level of CA before and after one or three muscle contractions, as well as after
 reattachment; baseline noise level of MEAP before and after ten muscle contractions;
 baseline noise change rates before and after muscle contractions; SNR performance of
 the last muscle contraction recorded by each of the CA and MEAP channels.

Significance was determined by one sample t test (*P< 0.05; **P< 0.01; ***P< 0.001;
****P< 0.0001).

We also added statistical information into the caption of Fig. 8 and 'Materials and
Methods' section.

See page 28:

... by 6 channels in column 1 of MEAP.

f RMS values of sEMG signals against time (left), in the isometric task (middle) and in
the dynamic task (right) across selected channels. n = 54 RMS values per channel. The
box plots show the mean (center square), median (center line), the 25th to 75th
percentiles (box) and the smallest and largest value that is ≤ 1.5 times the interquartile
range (the limits of the lower and upper whiskers, respectively)

see page 37:

**Statistical analysis:**

Data are presented with mean values \pm SD, unless otherwise noted in the figure
caption. Significance was defined as *P < 0.05; **P < 0.01; ***P < 0.001; ****P <
0.0001. Statistical analysis was performed using Origin Pro 2021.

We also have other statistical analysis in supplementary information, including
Supplementary Fig. 26-28. Moreover, to explore the potential applications of our MEAP,
we performed the same experiment on three subjects, and comparable results were
obtained. This evidence indicates that our MEAP can be applied consistently and
reliably across different individuals. We agree with the reviewer's suggestion that
including more statistical analysis in this section would better demonstrate the
advantages of our tools. However, as this manuscript primarily focuses on introducing
a new tool rather than presenting a specific clinical application, we believe that too
many samples or tests are not necessary to establish the novelty and validity of our tools.
It is important to consider that conducting statistical experiments would require

significant additional expenses and time. We have also observed that the level of
statistical analysis in our manuscript exceeds what is typically found in other similar
publications discussing novel electrodes¹³⁻¹⁵. Overall, while we recognize the value of
additional statistical analysis, we believe that the current evidence and results are
sufficient to establish the uniqueness and potential of our MEAP tool for further
exploration and development.

*As it stands, there is no evidence for the results in the different applications, and for me*
*that is a no-go for a publication.*

**Our response:**

We appreciate the reviewer's perspective on the applications of our work. However, we
would like to emphasize and clarify the comprehensiveness of our validations again. In
Figure 5, we have already provided evidence that the conformability of our TPP
electrode is superior to that of commercial Ag/AgCl electrodes on the forehead, and the
conformability of MEAP is significantly better than CA on the biceps brachii. Also, we
have shown that different attachment methods significantly impact the Signal-to-Noise
Ratio (SNR). Therefore, logically, repeating the same comparisons in our subsequent
applications is unnecessary, as the issue of conformability arises in most areas of the
body. It is worth noting that in subjects with lower body fat percentage, this issue can
be more pronounced due to greater skin deformation. However, with our MEAP, we
have observed excellent performance not only in signal quality but also in RMS
recording, fatigue recording, and tendon displacement. As a result, this tool is expected
to exhibit superior performance in most sEMG recordings where skin deformation
occurs (which is prevalent across the body). Furthermore, the successful observations
of RMS, fatigue, and tendon displacement enable the application of this tool in muscle
injury prevention. Because monitoring these parameters can help control tendon length,
which is crucial in preventing tendon tears—a common cause of injury. It is important
to clarify that our intention in this article is not to establish clinical criteria, but rather
to show the capabilities of this tool and its potential applications based on those

capabilities. However, we would be delighted to conduct further clinical studies with
statistical data using our tools in our future studies. Considering the aforementioned
characteristics, we firmly believe that MEAP brings innovation to the current tool
market, aligning with the standards of the journal.

*Furthermore, some of the claims about the state of the art of existing sEMG*
*electrode/lead methods are simply wrong. Let me cite the abstract: "However, current*
*sEMG electrodes do not offer adequate high-quality data for their widespread use in*
*clinics and everyday life, since these are neither stretchable nor arrayed". This sentence*
*is simply wrong! Electrode arrays have been available for many years and some of them*
*are stretchable. Whether stretchability improves the clinical significance of the sEMG*
*is unknown, while the clinical benefit of the array is proven. The manuscript is full of*
*such examples and should be formulated in a less absolute way to reflect reality.*

**Our response:**

We appreciate the reviewer comments on our inappropriate phrasing. We changed our
abstract as the reviewer advice.

See page 1:

**‘Surface electromyography (sEMG) can provide multiplexed information about muscle**
**performance. If current sEMG electrodes are stretchable, arrayed, and able to be used**
**multiple times, they would offer adequate high-quality data for continuous monitoring.**
**The lack of these properties delays the widespread use of sEMG in clinics and in**
**everyday life. Here, we address these constraints by design of an adhesive dry electrode**
**using tannic acid, polyvinyl alcohol, and PEDOT:PSS (TPP). The TPP electrode offers**
**superior stretchability (~200%) and adhesiveness (0.58 N/cm) compared to current**
**electrodes, ensuring stable and long-term contact with the skin for recording (>20**
**dB; >5 days). Additionally, we developed a metal-polymer electrode array patch**
**(MEAP) comprising liquid metal (LM) circuits and TPP electrodes. The MEAP**
**demonstrated better conformability than commercial arrays, resulting in higher signal-**

to-noise ratio and more stable recordings during muscle movements. Manufactured
using scalable screen-printing, these MEAPs feature a completely stretchable material
and array architecture, enabling real-time monitoring of muscle stress, fatigue, and
tendon displacement. Their potential to reduce muscle and tendon injuries and enhance
performance in daily exercise and professional sports holds great promise.'

To address the concern raised by the reviewer, we conducted a thorough examination
of our manuscript and made appropriate changes. We specifically focused on ensuring
that all the conclusions presented in the "Results" section were supported by our
experimental findings, thereby minimizing any subjective aspects. Consequently, we
have also revised the introduction and discussion sections to further enhance the clarity
and objectivity of our work.

See page 2: 'However, there is very little research using sEMG techniques to make such
tendon identifications.'

See page 29: 'However, it is extremely hard for commercial hydrogel electrodes to
accomplish the same recording sites in the same area as MEAP.'

With regard to reviewer's comments about the clinical significance of stretchability, we
agree that its direct clinical impact has never been examined. However, it is precisely
because of this uncertainty, that we are utilizing our new tool, which has demonstrated
clear advantages in terms of conformability and signal quality during movement, to
explore its potential applications in clinical scenarios. Throughout the manuscript, we
have utilized phrases such as 'we believe,' 'it would be,' 'had great potential,' and 'in
the future' to emphasize that we are not claiming immediate superiority over existing
clinical tools but rather aiming to facilitate further explorations based on our findings.

In summary, we greatly appreciate the reviewer's insightful comments as they have
significantly improved the manuscript.

**Reference**

- 1. Cheng, J. *et al.* Wet-Adhesive Elastomer for Liquid Metal-Based Conformal
Epidermal Electronics. *Adv Funct Mater* 2200444 (2022)
doi:10.1002/adfm.202200444.
- 2. Zhong, B., Jiang, K., Wang, L. & Shen, G. Wearable Sweat Loss Measuring Devices:
From the Role of Sweat Loss to Advanced Mechanisms and Designs. *Advanced*
*Science* **9**, (2022).
- 3. Fridén, J., Lovering, R. M. & Lieber, R. L. Fiber length variability within the flexor
carpi ulnaris and flexor carpi radialis muscles: Implications for surgical tendon
transfer. *Journal of Hand Surgery* **29**, 909–914 (2004).
- 4. Campos, D., Bartholdy, L. M. & Souza. *Case report Accessory flexor carpi ulnaris*
*muscle: a case report of a rare variation in human. J. Morphol. Sci* vol. 27 (2010).
- 5. Mesin, L., Merletti, R. & Rainoldi, A. Surface EMG: The issue of electrode location.
*Journal of Electromyography and Kinesiology* **19**, 719–726 (2009).
- 6. Martinez-Valdes, E. *et al.* Consensus for experimental design in electromyography
(CEDE) project: Single motor unit matrix. *Journal of Electromyography and*
*Kinesiology* **68**, 102726 (2023).
- 7. Gallina, A. *et al.* Consensus for experimental design in electromyography (CEDE)
project: High-density surface electromyography matrix. *Journal of*
*Electromyography and Kinesiology* **64**, 102656 (2022).
- 8. Besomi, M. *et al.* Consensus for experimental design in electromyography (CEDE)
project: Amplitude normalization matrix. *Journal of Electromyography and*
*Kinesiology* **53**, 102438 (2020).
- 9. Mcmanus, L. *et al.* Consensus for experimental design in electromyography (CEDE)
project: Terminology matrix. *Journal of Electromyography and Kinesiology* **59**,
102565 (2021).
- 10. Besomi, M. *et al.* Consensus for experimental design in electromyography (CEDE)
project: Electrode selection matrix. *Journal of Electromyography and Kinesiology*
**48**, 128–144 (2019).
- 11. Hermens, H. J., Freriks, B., Disselhorst-Klug, C. & Rau, G. *Development of*
*recommendations for SEMG sensors and sensor placement procedures. Journal of*
*Electromyography and Kinesiology* vol. 10 www.elsevier.com/locate/jelekin (2000).
- 12. Merletti, R. & Muceli, S. Tutorial. Surface EMG detection in space and time: Best
practices. *Journal of Electromyography and Kinesiology* **49**, 102363 (2019).
- 13. Zhang, L. *et al.* Fully organic compliant dry electrodes self-adhesive to skin for long-
term motion-robust epidermal biopotential monitoring. *Nat Commun* **11**, 4683
(2020).
- 14. Tan, P. *et al.* Solution-processable, soft, self-adhesive, and conductive polymer
composites for soft electronics. *Nat Commun* **13**, 358 (2022).
- 15. Driscoll, N. *et al.* MXene-infused bioelectronic interfaces for multiscale
electrophysiology and stimulation. *Sci Transl Med* **13**, eabf8629 (2021).

REVIEWER COMMENTS

Reviewer #2 (Remarks to the Author):

The authors have addressed my concerns in the revised manuscript. I would like to recommend its publication in Nature communications.

Reviewer #3 (Remarks to the Author):

I am tired of the discussion with the authors. Either they don't seem to understand my criticisms or, what I think is more likely, they can't address them. Two examples:

The authors use 3M RedDot 2223 and 2228 electrodes. According to the manufacturer, the RedDot 2223 has a diameter of 43.1 mm. When two electrodes are taped side by side for a bipolar lead, the smallest possible interelectrode distance (from centre to centre) is 43 mm. How is an interelectrode distance of 20 mm to be achieved with this? Even if the adhesive surface is reduced by cutting, it is difficult to maintain an interelectrode distance of 20 mm because the active electrode surface has a diameter of 16 mm. With the remaining 4 mm, sufficient adhesion cannot be achieved, which worsens the SNR. Long story short: Neither the 3M RedDot 2223 nor the 2228 meet the SENIAM standard.

Secondly: Statistic is now calculated via repetitions in individual subjects. This makes no sense at all, since the performance of the new electrode depends on where and how well it sticks. And on the other hand, it is known that the performance of different electrodes depends on the individual subject. This is due to the different skin resistance of different test persons. Even if it was tested on three subjects, that is far too few to reach a sustainable conclusion. For me, such arbitrariness does not belong in a scientific paper. The statement alone that a proper study is too time-consuming and expensive (which, by the way, is not an argument) already shows that the authors concede that a scientifically correct investigation of the question could well lead to different results.

1 **Response to reviewers for the manuscript (NCOMMS-22-46103C)**

2

*Reviewer #2 (Remarks to the Author):*

*The authors have addressed my concerns in the revised manuscript. I would like to*

*recommend its publication in Nature communications.*

**Our response:** We truly appreciate the reviewer taking the time to carefully read our

revised manuscript and provide such excellent feedback.

*Reviewer #3 (Remarks to the Author):*

*I am tired of the discussion with the authors. Either they don't seem to understand my*

*criticisms or, what I think is more likely, they can't address them. Two examples:*

*The authors use 3M RedDot 2223 and 2228 electrodes. According to the manufacturer,*
*the RedDot 2223 has a diameter of 43.1 mm. When two electrodes are taped side by*
*side for a bipolar lead, the smallest possible interelectrode distance (from centre to*
*centre) is 43 mm. How is an interelectrode distance of 20 mm to be achieved with this?*
*Even if the adhesive surface is reduced by cutting, it is difficult to maintain an*
*interelectrode distance of 20 mm because the active electrode surface has a diameter*
*of 16 mm. With the remaining 4 mm, sufficient adhesion cannot be achieved, which*
*worsens the SNR. Long story short: Neither the 3M RedDot 2223 nor the 2228 meet the*
*SENIAM standard.*

**Our response:** We value the reviewer's time spent reading our revised manuscript and
raise the concern about electrodes usage again. In Fig. 4e, we presented a photograph
illustrating our method to achieve an IED of 20 mm by cutting the adhesive surface. It
is important to note that both the active electrode surface and the surrounding substrate
exhibit sufficient adhesiveness. Thus, the adhesive properties of the electrodes in this
configuration should be adequate to ensure high-quality recording. Notably, no motion
artifacts were observed in the recording depicted in Fig. 4g. Building upon our
statements in last response letter regarding electrode selection, we maintain that the 3M
RedDot 2223 and 2228 electrodes align with the SENIAM or CEDE standards.

*Secondly: Statistic is now calculated via repetitions in individual subjects. This makes*
*no sense at all, since the performance of the new electrode depends on where and how*

*well it sticks. And on the other hand, it is known that the performance of different*
*electrodes depends on the individual subject. This is due to the different skin resistance*
*of different test persons. Even if it was tested on three subjects, that is far too few to*
*reach a sustainable conclusion. For me, such arbitrariness does not belong in a*
*scientific paper. The statement alone that a proper study is too time-consuming and*
*expensive (which, by the way, is not an argument) already shows that the authors*
*concede that a scientifically correct investigation of the question could well lead to*
*different results.*

**Our response:** To address the reviewer's concern, we modified Fig. 7, 8 and added
statistics.

[revised manuscript text omitted]

(control) is compared with others. Significance was determined by one sample t test
(* $P < 0.05$).

**g** RMS values of sEMG signals...

**j** A visual representation of the potential for muscle injury index, generated based on
the assessments made using the MEAP for isometric and dynamic task. The assessment
is presented as a unified model using the measures obtained from **g-i**. The loads were
classed as safe (< 3 kg), effective (3-5 kg) and vulnerable (>5 kg) based on the subject's
previous experience.

Some other changes:

We removed **Supplementary Fig. 31 Summary of muscle information of three**
**subjects** since its content has been demonstrated in Fig. 8.

To give more details, we added Data analysis section in Materials and Methods.

**Data analysis:**

All RMS, median frequency, and mean frequency values of the recorded sEMG signals
were computed for time steps of 0.125 seconds, unless specified otherwise. For
dynamic tasks in muscle-tendon junction location section, the mean frequency values
of sEMG data were initially smoothed using a Savitzky–Golay filter (with a frame
length of 21 and an order of 1); for real-time monitoring of dynamic tasks in the same
section, each set of values were determined first and then the means were generated and

plotted as mean \pm SD. In the section of injury prevention, the Gardner-Altman plot
was generated with a confidence level of 0.95 and a total of 5000 bootstrap samples.

We made adjustments to some sentence structures within our manuscript to enhance
clarity and conciseness, while ensuring that no conclusions have been altered.

For some examples,

see page 13: Due to the excellent flexibility and adhesiveness of electrode and substrate,
TPP electrodes can always make perfect attachment to the skin no matter if the skin is
compressed or stretched (Fig. 4a).

See page 14: To reduce errors caused by fatigue, we linear fitted the first 25 s of each
contraction to quantify the change Linear fittings were made for first 25 s of each
contraction, to quantify the outcome with less errors⁷³.

See page 21: We verified our MEAP-based findings with ultrasound images of biceps
distal tendon in a representative subject while the subject performed the isometric task
with load of 5 kg (Fig. 7a, Supplementary Video 5). The positional difference of muscle-
tendon junction was about 3.81 cm between flexion and extension confirmed using
ultrasound image (Fig. 7b).

See page 22: We also obtained the real-time displacement of Achilles tendon junction
accurately when switching from plantarflexion to dorsiflexion even in different subjects
(Fig. 7g). Once the junction movement range was identified, a MEAP with shorter IED
of 6 mm was used to further improve the precision of the location (Fig. 7h).

See page 27: As a result, such a high-possible injured circumstance is depicted as red

range in the muscle injury index. For other three ranges, each one should be determined
specifically by the exerciser under professional instructions. The MEAP successfully
provided information about muscle loading, muscle fatigue and tendon displacement of
the other subjects, which verified the stable and reliable recording using MEAP across
all individuals (Supplementary Fig. 29, 30).